

**Chemical characterization of fine particular matter in Changzhou, China**
**and source apportionment with offline aerosol mass spectrometry**
Zhaolian Ye[1,2], Jiashu Liu[1], Aijun Gu[1], Feifei Feng[1], Yuhai Liu[1], Chenglu Bi[1], Jianzhong
Xu[3], Ling Li[2], Hui Chen[2], Yanfang Chen[2], Liang Dai[2], Quanfa Zhou[1], Xinlei Ge[2,*]
[1]College of Chemistry and Environmental Engineering, Jiangsu University of
Technology, Changzhou 213001, China
[2]Jiangsu Key Laboratory of Atmospheric Environment Monitoring and Pollution
Control, Collaborative Innovation Center of Atmospheric Environment and Equipment
Technology, School of Environmental Sciences and Engineering, Nanjing University of
Information Science and Technology, Nanjing 210044, China
[3]State Key Laboratory of Cryospheric Sciences, Cold and Arid Regions Environmental
and Engineering Research Institute, Chinese Academy of Sciences, Lanzhou 730000,
China
*Corresponding author, Email: caxinra@163.com
Phone: +86-25-58731394
**Abstract:** Knowledge on aerosol chemistry in densely populated regions is critical for
reduction of air pollution, while such studies haven't been conducted in Changzhou, an
important manufacturing base and polluted city in the Yangtze River Delta (YRD),
China. This work, for the first time, performed a thorough chemical characterization on
the fine particular matter (PM$_{2.5}$) samples, collected during July 2015 to April 2016
across four seasons in Changzhou city. A suite of analytical techniques were employed
to characterize organic carbon/elemental carbon (OC/EC), water-soluble organic carbon
(WSOC), water-soluble inorganic ions (WSIIs), trace elements, and polycyclic aromatic
hydrocarbons (PAHs) in PM$_{2.5}$; in particular, an Aerodyne soot particle aerosol mass
spectrometer (SP-AMS) was deployed to probe the chemical properties of water-soluble
organic aerosols (WSOA). The average PM$_{2.5}$ concentrations were found to be 108.3 μg



$m^{-3}$, and all identified species were able to reconstruct ~80% of the $PM_{2.5}$ mass. The
WSIIs occupied about half of the $PM_{2.5}$ mass (~52.1%), with $SO_4^{2-}$, $NO_3^-$ and $NH_4^+$ as
the major ions. On average, nitrate concentrations dominated over sulfate (mass ratio of
1.21), indicating influences from traffic emissions. OC and EC correlated well with each
other and the highest OC/EC ratio (5.16) occurred in winter, suggesting complex OC
sources likely including both secondarily formed and primarily emitted OA.
Concentrations of eight trace elements (Mn, Zn, Al, B, Cr, Cu, Fe, Pb) can contribute up
to 6.0% of $PM_{2.5}$ during winter. PAHs concentrations were also high in winter (140.25
ng $m^{-3}$), which were predominated by median/high molecular weight PAHs with 5- and
6-rings. The organic matter including both water-soluble and water-insoluble species
occupied ~20% $PM_{2.5}$ mass. SP-AMS determined that the WSOA had an average atomic
oxygen-to-carbon (O/C), hydrogen-to-carbon (H/C), nitrogen-to-carbon (N/C) and
organic matter-to-organic carbon (OM/OC) ratios of 0.36, 1.54, 0.11, and 1.74,
respectively. Source apportionment of WSOA further identified two secondary OA
(SOA) factors (a less oxidized and a more oxidized OA) and two primary OA (POA)
factors (a nitrogen enriched hydrocarbon-like traffic OA and a cooking-related OA). On
average, the POA contribution overweighed SOA (55% vs. 45%), indicating the
important role of local anthropogenic emissions to the aerosol pollution in Changzhou.
Our measurement also shows the abundance of organic nitrogen species in WSOA, and
the source analyses suggest these species likely associated with traffic emissions, which
warrants more investigations on PM samples from other locations.

**1. Introduction**
Aerosol particles are ubiquitous in the atmosphere and play important roles in air
quality, global climate, biogeochemical cycle, and human health, etc (e.g., Heal et al.,
2012;Cao et al., 2012;Hu et al., 2015). Aerosol pollution can also influence remote
territories via long-range transport. Therefore, atmospheric aerosol has received
extensive attentions from the government, public and academia (e.g., Zhang et al.,
2007;Jimenez et al., 2009). Particularly, much attentions have been focused on fine





particles ($PM_{2.5}$, aerodynamic diameters less than 2.5 µm) as they can go deeper into the
respiratory system, causing more severe health problems than coarse particles (Anderson
et al., 2012). However, as is well known, the concentrations, sources, chemical
compositions and formation mechanisms of $PM_{2.5}$ are complicated and can vary greatly
with meteorological conditions, seasons and regional/local topography, etc. $PM_{2.5}$ can
contain a variety of species, i.e., organic carbon/elemental carbon (OC/EC), trace
elements, inorganic salts, and various organic species such as polycyclic aromatic
hydrocarbons (PAHs)(e.g., Wang et al., 2015). In China, haze pollution occurred
frequently in recent years, and a large number of studies regarding the chemical
characterization of fine particles were carried out in many locations (Wang et al., 2006a),
such as Shanghai (e.g., Wang et al., 2016a;Zhao et al., 2015), Beijing (e.g., Sun et al.,
2014;Hu et al., 2016;Sun et al., 2016), Nanjing (e.g., Zhang et al., 2016;Ding et al.,
2013), Lanzhou (e.g., Fan et al., 2014;Xu et al., 2014), Wuhan (e.g., Huang et al., 2016),
and other remote sites (Xu et al., 2015), etc.

Yangtze River Delta (YRD) region, located in East China, is experiencing severe

atmospheric pollution along with the rapid economic development. Some studies carried
out in the YRD investigated different characteristics of the fine aerosols, including the
mass loading, composition, hygroscopicity (e.g., Ye et al., 2011;Ge et al., 2015), size
distribution, seasonal variation and source, formation pathway, and their impacts on
visibility and climate (e.g., Wang et al., 2012). However, these studies were mostly
limited in Nanjing (e.g., Hu et al., 2012;Wang et al., 2016b) and Shanghai (e.g., Fu et al.,
2012;Qiao et al., 2015;Wang et al., 2012). Changzhou, situated in the western YRD
region, between Shanghai and Nanjing, is also a major city and an important
manufacturing base due to its geographical advantage. The city has an area of about
4374 $km^2$ with a population of 4.45 million. Due to elevated emissions of various
pollutants, the number of hazy days increased over the past few years in Changzhou as
well. To the best of our knowledge, no work has been published specifically on chemical
characteristics and source apportionment of fine particles in Changzhou. Thus, it is
scientifically and practically important to investigate the $PM_{2.5}$ characteristics in order to



provide efficient control strategies to reduce the PM pollution in Changzhou.
Among various PM$_{2.5}$ constituents, organic aerosol (OA) is a vital component,
accounting for a significant, even dominant fraction of PM$_{2.5}$ in ambient air (Zhang et al.,
2007). Thus elucidation of its constituents, properties and sources is essential.
Apportionment of OA into different sources correctly is a critical step towards enabling
efficient air pollution control strategies. Recently, Aerodyne Aerosol Mass spectrometry
(AMS) has been used extensively for quantitatively characterizing ambient OA, and the
obtained wealthy mass spectral data allows a better source analyses of OA (Canagaratna
et al., 2007). Particularly, positive matrix factorization (PMF), as a standard multivariate
factor analysis method, has been widely applied on AMS data sets to distinguish and
quantify the OA sources (Zhang et al., 2011). Many previous studies (e.g., Ge et al.,
2012a;Ng et al., 2011) have deployed the AMS for online field measurements since
AMS can provide real-time information on mass concentrations and size distributions of
aerosol particles with very fine time resolution (~several minutes). However, up to now,
AMS was typically used for short-term online measurement and only a few studies made
efforts to apply it on offline filter samples analyses and source apportionment (Ge et al.,
2014;Daellenbach et al., 2016;Sun et al., 2011a).
In this study, for the first time, we systematically investigated the chemical
characteristics of ambient PM$_{2.5}$ collected in Changzhou nearly across one-year period,
providing an overview about the concentrations of PM$_{2.5}$, water-soluble inorganic ions
(WSIIs), trace elements, carbonaceous species, water-soluble organic carbon (WSOC),
and PAHs in PM$_{2.5}$, and the relationships among these components. Seasonal variations
of different PM$_{2.5}$ components were also discussed. Further, we employed an Aerodyne
soot particle aerosol mass spectrometer (SP-AMS) (Onasch et al., 2012;Lee et al.,
2015;Wang et al., 2016c) to investigate the properties and potential sources of OA on
the basis of high resolution mass spectra determined by the SP-AMS. Findings from this
study also adds knowledge to the framework of Pan-Eurasian Experiment (PEEX)
(Kulmala et al., 2015).
**2. Experiments**



## 2.1. Sampling site and PM$_{2.5}$ collection


The sampling site was set on the rooftop of a nine-story building inside the campus of
Jiangsu University of Technology in Changzhou (31.7$^{o}$N, 119.9$^{o}$E), as shown in Fig. 1.
This site locates in the southwestern part of Changzhou, surrounded by a residential area,
approximately 0.5 km away from an urban street - Zhongwu Road, and has no direct
influences from industrial emissions (14.7 km away from the closest industrial plant–
Bao Steel). Meteorological parameters including temperature, relative humidity (RH),
wind speed (WS), wind direction (WD), and concentrations of gas-phase species such as
SO$_2$ and NO$_2$ are recorded by the air quality monitoring station inside the campus, which
is about 500 m from the sampling site. Average meteorological parameters of four
seasons are shown in Table 1. The wind rose plots of different seasons are shown in Fig.
S1 in the supplement. The wind speed was generally low in Changzhou (on average, 1.1,
1.6, 0.9 and 0.8 m s$^{-1}$ in spring, summer, fall and winter, respectively).
PM$_{2.5}$ were collected onto 90 mm quartz fiber filters (Whatman, QM-A) using a
medium volume sampler (TH-150 C, Wuhan Tianhong Ltd., China) with a flow rate of
100 L min$^{-1}$. The filters, wrapped in aluminum foil, were prebaked at 450 °C for 4 h
prior to sampling. The sampler began to collect particles at 9:00 am and stopped at 5:00
am in the following day, ensuring the duration time for each sample of 20 h. A total of
69 PM$_{2.5}$ samples were collected in 2015-2016: 20 July - 19 August 2015 (summer, 11
samples), 18 September - 25 October 2015 (fall, 23 samples), 7 December 2015-15
January 2016 (winter, 24 samples) and 1 March -12 April 2016 (spring, 11 samples).
Before and after sampling, the filters were conditioned under constant temperature
(22±1°C) and relative humidity (45±5%) for 48 h and weighted by a microbalance
(precision of 0.01 mg). The filters were then wrapped and sealed in aluminum foil
envelopes separately, stored in a freezer at −20 °C until analysis to minimize the
evaporation loss of volatile components.
**2.2 Chemical analysis**
**2.2.1 IC analysis**
One quarter of a filter was put into a glass tube and 25 mL deionized water (18.2



M$\Omega$ cm$^{-1}$) was then added. After 15 min ultrasonic extraction, the solution was filtrated
through an acetate-cellulose filter with 0.45 μm pore size. Concentrations of the WSIIs
in the aqueous extract, including five anions (F$^-$, Cl$^-$, NO$_2^-$, NO$_3^-$, SO$_4^{2-}$) and five
cations (Na$^+$, NH$_4^+$, K$^+$, Mg$^{2+}$, Ca$^{2+}$), were then measured by the ion chromatograph (IC,
Dionex ICS-600 for anions and ICS-1500 for cations). The method detection limits
(MDL) were determined to be 18.0, 7.3, 5.2, 6.3, 11.0, 18.7, 3.3, 4.6, 2.6, and 11.5 μg
L$^{-1}$ for F$^-$, Cl$^-$, NO$_2^-$, NO$_3^-$, SO$_4^{2-}$, Na$^+$, NH$_4^+$, K$^+$, Mg$^{2+}$ and Ca$^{2+}$, respectively, and all
measured concentrations were above the MDLs. Note the filter blanks were treated in
the same way, and all data for the samples reported here were blank corrected, other
analyses in the following sections were also blank corrected unless specified. The
concentrations of all measured species in PM$_{2.5}$ sample were also converted to μg m$^{-3}$
based on the measured concentrations and the air volume pulled through the filter.
**2.2.2 ICP-OES analysis**
Another quarter of a filter was cut and placed in a Teflon vessel, digested with 10
mL mixture of HNO$_3$-HCl (1:1, v:v) in a microwave system (XT-9900A, Shanghai
Xintuo Co.) for 8 h. After the digested solution cooled down to room temperature, it was
filtered through a 0.45 μm acetate-cellulose filter. The filtrate was then diluted using
deionized water to 50 mL, and analyzed using Optima 8000 (Perkin Elmer, USA)
inductively coupled plasma atomic emission spectrometry (ICP-OES) to determine
concentrations of eight trace elements (Mn, Zn, Al, B, Cr, Cu, Fe, Pb). It is worth to
mention that we also tried to measure the concentrations of other trace elements such as
Ti, Ni, Ba, but found they were mostly below the detection limits thus were not included
in this work. All samples were determined in a triplicate, and a difference within 5%
was considered acceptable.
**2.2.3 OC/EC and WSOC analysis**
Analysis procedure of OC/EC was similar to a previous study (Zhao et al., 2015) .
Briefly, OC and EC were measured by the DRI model 2001 thermal/optical carbon
analyzer (Atmoslytic Inc. Calabasas, CA) using a 0.526 cm$^2$ punch from each filter,
following the IMPROVE TOR protocol (Chow et al., 2004). Filter was measured



stepwise at temperatures of 140 °C ($OC_1$), 280 °C ($OC_2$), 480 °C ($OC_3$), and 580 °C
($OC_4$) in a helium atmosphere, and 580 °C ($EC_1$), 740 °C ($EC_2$), and 840 °C ($EC_3$) in a 2%
oxygen/98% helium gas atmosphere. OC is calculated as $OC_1+OC_2+OC_3+OC_4+OP$ and
EC as $EC_1+EC_2+EC_3-OP$, where OP is the optical pyrolyzed OC.
The WSOC concentrations were determined by a TOC analyzer (TOC-L, Shimazu,
Japan). Instrument details and procedure of the WSOC analysis can be found in our
previous work (Ge et al., 2014).

**183    2.2.4 GC-MS analysis for PAHs**

Due to the limitation of samples, we only analyzed PAHs for spring and winter. The
PAHs analysis was conducted following the standard procedure, similar to the work of
Szabó et al. (2015). One quarter of a filter was treated by Soxhelt extraction for 18 h
using 250 mL mixture of *n*-hexane/ethylether (5:1, v/v). To determine the recovery rates,
100 ng of deuterated surrogate standard solution containing naphthalene-$d_8$ and
perylene-$d_{12}$ (o2si, USA) was added into the sample prior to extraction, and the average
recovery rates of $d_8$ and $d_{12}$ were over 90%. The extracts were then concentrated to
about 2 mL by a rotary evaporator, purified in a chromatography column (filled with 3
cm deactivated $Al_2O_3$, 10g silica gel, 2 cm deactivated $Na_2SO_4$). The column was first
eluted with 25 mL *n*-hexane and the eluate was discarded, then elution was carried out
using 30 mL dichloromethane/*n*-hexane (1:1,v:v). Samples containing PAHs were again
concentrated to about 2 mL by the rotary evaporation. Finally they were condensed to
exactly 1 mL under a gentle $N_2$ stream in a 60 °C water bath. The extracts are transferred
into ampoule bottles and stored in a refrigerator until analysis.
The PAH compounds in the final extracts were analyzed with a gas
chromatography - mass spectrometer (GC-MS) (Agilent 7890-7000B, USA), using a
DB-5ms capillary column (30 m×0.25 mm×0.5 μm). The instrument conditions were
set as follows: injector at 200 °C; ion source at 230 °C; the column was programmed at
40 °C for 2 min, then increased to 100 °C at a rate of 10 °C min$^{-1}$, held for 1 min, then
increased to 250 °C at 20 °C min$^{-1}$, and finally held for 3 min at 250 °C. The mass
selective detector was operated in the electron impact mode using 70 eV. Multi reaction





monitor modes were employed for the identification and quantification of PAHs.
Before sample analysis, calibration standards at a series of concentrations were
prepared from aromatic hydrocarbon standard (O2si, USA) containing 18 PAH
compounds (1000 mg L$^{-1}$), which are naphthalene (NaP) ($C_{10}H_8$), acenaphthylene (Acy)
($C_{12}H_8$), acenaphthene (Ace) ($C_{12}H_{10}$), fluorene (Flu) ($C_{13}H_{10}$), phenanthrene (Phe)
($C_{14}H_{10}$), anthracene (Ant) ($C_{14}H_{10}$), fluoranthene (Flua) ($C_{16}H_{10}$), pyrene (Pyr) ($C_{16}H_{10}$),
benzo(a)anthracene (BaA) ($C_{18}H_{12}$), chrysene (Chr) ($C_{18}H_{12}$), benzo(b)fluoranthene
(BbF) ($C_{20}H_{12}$), benzo(k)fluoranthene (BkF) ($C_{20}H_{12}$), benzo(a)pyrene(BaP) ($C_{20}H_{12}$),
Benzo(e)pyrene (BeP) ($C_{20}H_{12}$), benzo(j)fluoranthene (BjF) ($C_{20}H_{12}$),
benzo(ghi)perylene (BghiP) ($C_{22}H_{12}$), indeno(1,2,3-cd)pyrene (InP) ($C_{22}H_{12}$), and
dibenz(a,h)anthracene (DBA) ($C_{22}H_{14}$). These PAHs can be classified by the number of
aromatic rings and molecular weights: low molecular weight (LMW) PAHs containing
2- and 3-rings (NaP, Acy, Ace, Flu, Phe, Ant), medium molecular weight (MMW)
PAHs containing 4-rings (Flua, Pyr, BaA, Chr) and high molecular weight (HMW)
PAHs containing 5- and 6-rings (BbF, BkF, BjF, BaP, BeP, InP, DBA, BghiP) (Wang et
al., 2015;Kong et al., 2015). The calibration was conducted twice prior to analysis.
Identification and quantification of each PAH is based on its retention time and peak
areas in the calibration curve and sample curve, and the total PAH concentration (Σ
PAH) was calculated as the sum of concentrations of all 18 individual PAHs. Figure S2
shows examples of the GC-MS spectra of a few 18-PAHs standards and two surrogate
standards ($d_8$ and $d_{12}$).
**2.2.5 Offline SP-AMS analysis**
The SP-AMS analysis procedure for offline filters was similar to that of Xu et al.
(2013). Briefly, for each sample, 1/4 filter was extracted in 25 mL deionized water. The
liquid extracts were aerosolized using an atomizer (TSI, Model 3076), and the mist
passed through a silica-gel diffusion dryer, leaving dry particles which were
subsequently analyzed by the SP-AMS. Note the SP-AMS was operated with the laser
off so similar to other AMS measurements; it measured non-refractory organic species
that can vaporize fast at the oven temperature of 600 $^{\circ}$C. The instrument employs the 70





eV electron impact (EI) ion generation scheme, all vaporized species were broken into
ion fragments with specific mass-to-charge ($m/z$) ratios, and the time-of-flight mass
spectrometer outputs the mass spectrum that records the ions according to their signal
intensities and $m/z$ ratios. Ion fragments with $m/z$ up to 300 amu were recorded in this
study. The SP-AMS mass spectra can well represent the total OA constituents, and the
bulk OA properties such as elemental ratios including oxygen-to-carbon (O/C),
hydrogen-to-carbon (H/C) and nitrogen-to-carbon (N/C) ratios, and the organic
mass-to-organic carbon (OM/OC) ratio can be obtained. Note although the SP-AMS is
limited in molecular-level speciation analysis (Drewnick, 2012), some compounds can
be identified via recognition of the fingerprint ions, and particular sources can be
separated and quantified via further factor analyses.

The SP-AMS data were processed using the Igor-based software toolkit

SQUIRREL (version 1.51H) and PIKA (version 1.10H) (downloaded from:
http://cires.colorado.edu/jimenez-group/ToFAMSResources/ToFSoftware/index.html),
and the analysis procedure was similar to our previous work (Ge et al., 2012b). We did
some minor modifications on the fragment table. For example, we set the organic $CO_2^+$
signal equal to organic $CO^+$, as the $CO_2^+$ signal in $PM_{2.5}$ may come from carbonate not
organics, and since we used Argon as carrier gas so different from ambient
measurements, the $CO^+$ signal can be well separated and quantified from $N_2^+$ at $m/z$ 28
(example shown in Fig. S3). Accordingly, organic $H_2O^+$, $HO^+$, $O^+$ were scaled to $CO_2^+$
using the ratios proposed by Aiken et al. (2008), and the elemental compositions and
H/C, N/C, O/C and OM/OC ratios of OA reported in this study were also determined
according to the method of Aiken et al. (2008).
**2.3 Determination of WSOA, WIOA**

Mass concentration of water-soluble organic mass (WSOA) were calculated by

multiplying the WSOC concentrations determined from the TOC analyzer with the
OM/OC ratios calculated from the SP-AMS mass spectra (Fig. 2) (equation 1). As
shown in Fig. 2, most OM/OC values were within the range of 1.4-2.1, in consistent
with the typical OM/OC ratios observed at other urban sites.





The water-insoluble organic carbon (WIOC) mass was calculated as the difference
between the OC determined by the OC/EC analyzer and the WSOC, and a factor of 1.3
suggested by Sun et al. (2011a), was used to convert WIOC mass to the mass of
water-insoluble organic matter (WIOA) (equation 2). The total organic matter (OA) was
treated as the sum of WSOA and WIOA (equation 3).
$$\text{WSOA} = \text{WSOC} \times OM/OC_{WSOA} \qquad (1)$$
$$\text{WIOA} = (\text{OC}-\text{WSOC})*1.3 \qquad (2)$$
$$\text{OA} = \text{WSOA}+\text{WIOA} \qquad (3)$$
**2.4 Source apportionment of WSOA**
In this work, we used the PMF Evaluation Toolkit v 2.06 (Ulbrich et al., 2009) and
followed the protocol described by Zhang et al. (2011) to conduct the PMF analyses.
Prior to PMF execution, the following steps were performed: Data and error matrix for
WSOA were first adjusted based on equation 1; ions with low signal-to-noise (S/N<0.2)
were removed, whereas ions with S/N ratios between 0.2 and 2 were downweighted;
Two runs with huge mass loading spikes were removed; all isotopic ions were removed
since their signals are not measured directly but scaled to their parent ions. The PMF
solutions were explored by varying the factors from 1 to 8 and the rotational forcing
parameter (*f*peak) from −1 to 1 with an increment of 0.1. The four-factor solution with
*f*peak=0 was chosen as the best solution in this study. The mass spectra of three-factor
and five-factor solutions were presented in Fig. S4. The three-factor solution does not
resolve well the oxygenated OA factors as many oxygenated ions were mixed with the
primary OA factors. The five-factor solution splits the cooking-related OA into two
similar factors based on the spectral patterns. Also, by investigating the correlations of
the factors with their corresponding tracer ions, and sulfate, nitrate, etc., of the 3-, 4-,
and 5-factor solutions, the 4-factor solution was found to be the most reliable and
representative solution.

**3. Results and discussion**
**3.1 Overview of PM$_{2.5}$ concentrations and components**



The annual and seasonal average concentrations of PM$_{2.5}$, OC, EC, OA, WSIIs,
trace elements and PAHs are summarized in Table 2. As shown in Table 2, the PM$_{2.5}$
concentrations (in μg m$^{-3}$) were on average (±1σ) 106.0 (±24.4), 80.9 (±37.7), 103.3
(±28.2), and 126.9 (±50.4) in spring, summer, fall and winter, respectively, with annual
average of 108.3 (±40.8), comparable to the PM$_{2.5}$ concentrations in Nanjing (106 μg
m$^{-3}$ in 2011) (Shen et al., 2014), Tianjin (109.8 μg m$^{-3}$ in 2008) (Gu et al., 2010) and
Hangzhou (108.2 μg m$^{-3}$ in 2004-2005) (Liu et al., 2015), but lower than that in Jinan
(169 μg m$^{-3}$ in 2010) (Gu et al., 2014). The PM$_{2.5}$ concentrations were highest in winter
and relatively low in summer, similar to those found in most cities, such as Tianjin (Gu
et al., 2010) and Hangzhou (Liu et al., 2015). Previous studies shows that low
concentrations occurring in summer are mainly due to the relatively high boundary layer
height, low RH and high temperature (Cheng et al., 2015;Huang et al., 2010). The
temperatures and RH values were on average 32.1℃ and 61.1% in summer during the
observation period (Table 1). Overall, the daily average concentration of PM$_{2.5}$ during
sampling period exceeds 75 μg m$^{-3}$ - the second-grade national air quality standard
(NAAQS)(GB 3095-2012), and on some heavily polluted days, the PM$_{2.5}$ mass loadings
can even exceed 3 times the NAAQS standard.
Table 2 summarizes the concentrations of various species determined in this study.
Overall, the reconstructed PM$_{2.5}$ mass estimated by the sum of OA, EC and WSIIs vs
gravimetrically determined PM$_{2.5}$ mass were shown in Fig. 3(a-d). The mass proportions
of all measured components to the PM$_{2.5}$ mass are illustrated by five inserted pie charts
representing four seasons and the whole year, respectively. On average, the quantified
species can occupy 77.3% of the PM$_{2.5}$ mass (note trace elements were not included as
they were only determined for spring and winter samples), and the mass closure appears
to be better for spring and winter samples. Overall, our results are similar to some
previous results, such as in Beijing (68%) (Zhang et al., 2013). Details and
characteristics of individual components are discussed in the following sections.

**3.2 Water soluble inorganic ions**

The average concentrations (±σ) of total WSIIs were 66.5 (±17.2), 35.0 (±20.2),





51.0 (±17.2), and 66.8 (±23.6) µg m$^{-3}$ in spring, summer, fall and winter, respectively,
with an annual average of 56.4 (±22.9) µg m$^{-3}$. The level was lowest in summer likely
due to the conditions favorable for pollutants dispersion and the wet scavenging on these
ions under summer monsoon circulation and precipitation. In total, all WSIIs can
account for 62.6%, 41.1%, 49.0% and 50.4% of PM$_{2.5}$ mass in spring, summer, fall and
winter, respectively, with the annual average WSIIs/PM$_{2.5}$ ratio of 52.1%, a little higher
than previously reported value of 45.3% in Handan in 2013 (Meng et al., 2016).
The mass fractions of ions to total WSIIs followed the order: NO$_3^-$ (34.2%)> SO$_4^{2-}$
(31.0%)> NH$_4^+$(21.2%)> Cl$^-$(6.0%)> Na$^+$(3.8%)> K$^+$(1.8%)> Ca$^{2+}$(1.2%)>
Mg$^{2+}$(0.3%) >NO$_2^-$ and F$^-$(0.2%) (Fig. 4b). Secondary inorganic ions including SO$_4^{2-}$,
NO$_3^-$, and NH$_4^+$, constitute the majority of total WSIIs (86.4%) (Fig. 4b) with the
highest one being NO$_3^-$. Nitrate and ammonium concentrations displayed distinct
seasonal variations - highest in spring (NO$_3^-$: 26.4 µg m$^{-3}$, NH$_4^+$: 14.8 µg m$^{-3}$), following
by winter (24.1 and 13.1 µg m$^{-3}$), and lowest in summer (6.8 and 8.2 µg m$^{-3}$). On the
other hand, as a non-volatile species, sulfate concentrations showed no obvious seasonal
differences.
The cross-correlation relationships between different ions can be used to infer their
possible common sources. Figure 5 shows the Pearson's correlation coefficients ($r$)
between ions for four seasons, respectively. As illustrated, NH$_4^+$ had good correlations
with SO$_4^{2-}$ and NO$_3^-$ ($r$>0.70), and particularly high $r$ values were found in winter (with
SO$_4^{2-}$: $r$=0.90, with NO$_3^-$: $r$=0.96) and summer (with SO$_4^{2-}$: $r$=0.98, with NO$_3^-$: $r$=0.93),
indicating these three ions were mainly present in the form of ammonium nitrate and
ammonium sulfate and were all formed secondarily. Moreover, the correlations between
Na$^+$ and Cl$^-$ varied largely with the seasons, poor in summer ($r$=-0.192) and winter
($r$=0.37), indicating different sources for them. For chloride, the annual average Cl$^-$/Na$^+$
mass ratio was 1.58, larger than 1.17 in seawater (Zhang et al., 2013), indicating the
important contributions from anthropogenic activities to chloride (such as coal
combustion) in Changzhou, in particular in winter as the content of Cl$^-$ in winter was
significantly elevated. By contrast, K$^+$ and Cl$^-$ have good correlations ($r$ of 0.86, 0.76,





0.80 and 0.62 in spring, summer, fall and winter), suggesting that $K^+$ may co-emit with
chloride. According to correlation analysis in Fig. 5, $Mg^{2+}$ and $Ca^{2+}$ had good relations
with $r$ of 0.58, 0.80, 0.81 and 0.78 in spring, summer, fall and winter, respectively,
indicating a similar source likely crustal material for these two ions.

Acidity of $PM_{2.5}$ can be evaluated by AE (anion equivalence) vs. CE (cation

equivalence), which is calculated by converting the concentrations of anions and cations
($\mu g\ m^{-3}$) into molar concentrations ($\mu mol\ m^{-3}$) using the following equations.
$$AE = \frac{SO_4^{2-}}{48} + \frac{NO_3^-}{62} + \frac{NO_2^-}{46} + \frac{Cl^-}{35.5} + \frac{F^-}{19} \qquad (4)$$

$$CE = \frac{NH_4^+}{18} + \frac{Mg^{2+}}{12.2} + \frac{Ca^{2+}}{20} + \frac{K^+}{39} + \frac{Na^+}{23} \qquad (5)$$


Figure 6a illustrates the scatter plots of CE vs. AE in four seasons. The slopes were 1.18,
1.09, 1.03 and 0.93 in spring, summer, fall and winter, respectively, indicating the
particles are generally neutralized. Normally, the ratio of $NH_4^+{}_{meas}/\ NH_4^+{}_{pred}$, proposed
by Young et al. (2016), can be used to evaluate the existing form of $NH_4^+$ ion. The
predicted $NH_4^+$ ($NH_4^+{}_{pred}$) was calculated using Equation 6.
$$NH_{4\ pred}^+ = 18 \times (2 \times \frac{SO_4^{2-}}{96} + \frac{NO_3^-}{62} + \frac{Cl^-}{35.5}) \qquad (6)$$

Figure S5 illustrated the ratio of $NH_{4meas}^+/NH_4^+{}_{pred}$ in $PM_{2.5}$ during four seasons. As
presented, the ratios were 0.95, 0.93, 0.87, 0.75 in spring, summer, fall and winter,
respectively, indicating that $(NH_4)_2SO_4$ and $NH_4NO_3$, $NH_4Cl$ were dominant forms for
these ionic species. However, the ratio in winter was only 0.75, much less than 1,
revealed that the ionic components of $PM_{2.5}$ in winter were more complicated than those
in other seasons, reflecting the probability that $PM_{2.5}$ contains other ions such as organic
cations in winter.

In addition, the mass ratio of $NO_3^-$ to $SO_4^{2-}$ ($NO_3^-/SO_4^{2-}$) can be used to identify

whether mobile sources (vehicle) or stationary sources (coal combustion) are dominant
for these ions (Wang et al., 2006b;Arimoto et al., 1996). When the $NO_3^-/SO_4^{2-}$ mass
ratio exceeds 1, it means that particle sources at the observation site are dominated by
mobile sources, while fixed sources play major roles when the ratio is below 1. In this





study, the mass ratios of $NO_3^-/SO_4^{2-}$ in sampling site were 1.52, 0.43, 0.99 and 1.29 in
the spring, summer, fall and winter, respectively, with an annual average ratio of 1.21
(Fig. 6b). The $NO_3^-/SO_4^{2-}$ ratio varied largely with seasons. Note in summer, a lower
$NO_3^-/SO_4^{2-}$ ratio may be also ascribed to high temperature which leads to the
dissociation of $NH_4NO_3$, yet the high $NO_3^-/SO_4^{2-}$ in winter and spring is more likely
relevant to traffic emissions from Zhongwu Road near the sampling site (Fig. 1).
Previous studies (Xu et al., 2014) have indicated that nitrogen oxidation ratio
(NOR= $nNO_3^-/(nNO_3^- + nNO_2)$, $n$ refers to the molar concentration), and sulfur oxidation
ratio (SOR= $nSO_4^{2-}/(nSO_4^{2-} + nSO_2)$), can be used to estimate the transformation of $NO_2$
and $SO_2$ to particle-phase $NO_3^-$ and $SO_4^{2-}$. The larger SOR and NOR mean more
secondarily formed nitrate and sulfate. The seasonal values for SOR and NOR are
plotted in Fig. 6 (c-d). The SOR appears to be higher in summer, indicating strong
photochemical oxidation for sulfate formation, while NOR is relatively higher in spring,
suggesting conversion of $NO_x$ into nitrate is more efficient in spring in Changzhou.
**3.3 Trace elements**
Eight trace elements (Mn, Zn, Al, B, Cr, Cu, Fe, Pb) of the samples collected
during fall and winter were determined in this study. The average concentrations ($\mu g\ m^{-3}$)
are shown in Fig. 7a. The total concentrations were 6.38 $\mu g\ m^{-3}$ and 2.77 $\mu g\ m^{-3}$,
accounting for 6.0% and 3.0% of the total $PM_{2.5}$ mass in winter and fall, respectively.
These values were relatively higher than those in other cities in China, such as
1.74%-2.04% in Hangzhou (Liu et al., 2015). This probably can be explained by
re-suspended dust from building construction around the site during the sampling period.
In this study, the observed mean levels of trace elements in fall were in the order of
Fe>Zn>B>Al>Cu>Mn>Pb>Cr, and ranked in Zn>Fe>B>Al>Cu>Mn>Pb>Cr in winter,
as demonstrated in Fig. 7a. In fall , Fe accounted for 39.0% of the total trace metal mass,
following by Zn (25.6%), B (12.3%) and Al (9.2%), while in winter Zn contributed the
largest (53.7%), following by Fe and B. Overall, Fe and Zn were the two most abundant
trace elements in $PM_{2.5}$, accounting for over half of the total trace elements mass.
Previous research also found that mass loading of Zn was higher than other elements,





even higher than Al in Nanjing in 2013 (Qi et al., 2016). Vehicle exhaust is likely one
major contributor to the high concentrations of Zn.
In general, the correlations between various heavy metals are weak, as depicted in
Fig. 7b-d, indicating that the complex sources including both natural and anthropogenic
sources for the trace metals observed here. For instance, Cr, Cu, Pb, and Zn can be
released from lubricating oils, tail pipe emissions, brake and tire wears (Zhang et al.,
2013); Fe and Mg are primarily crustal elements, while Zn and Cu are primarily from
anthropogenic sources. Fe and Al were only moderately correlated (for example, in fall
with $r$=0.74, Fig. 7b) showing that they are not from exactly same sources.

**3.4 OC and EC**

As presented in Table 2, the annual average EC concentration in Changzhou was
5.4 $\mu g\ m^{-3}$, close to Nanjing (5.3 $\mu g\ m^{-3}$) (Li et al., 2015) and Tianjin (5.9 $\mu g\ m^{-3}$)(Gu et
al., 2010), but lower than those in other cities (e.g., 22.3 $\mu g\ m^{-3}$ in Beijing (Duan et al.,
2012), and higher than that observed in Shanghai (2.8 $\mu g\ m^{-3}$)(Feng et al., 2009). The
seasonally averaged OC concentrations were highest in winter (18.3 $\mu g\ m^{-3}$), followed
by fall (13.2 $\mu g\ m^{-3}$) and spring (11.2 $\mu g\ m^{-3}$), and lowest in summer (7.9 $\mu g\ m^{-3}$). The
annual average OC concentration was 13.8 $\mu g\ m^{-3}$, comparable to those measured in
other cities, such as Shanghai (14.7 $\mu g\ m^{-3}$)(Feng et al., 2009), and Tianjin (16.9 $\mu g\ m^{-3}$)
(Gu et al., 2010).
The mass concentrations of total carbon (TC, the sum of OC and EC) were 16.0,
12.1, 21.0, 22.3 $\mu g\ m^{-3}$ in spring, summer, fall and winter, respectively (Table 2),
corresponding mass contributions to $PM_{2.5}$ were 15.3%, 17.5%, 19.7%, and 20.1% with
an annual mean of 18.1%. This value was similar to those measured in other cities in
China, such as Jinan (10-15%)(Gu et al., 2014), Shanghai (15%) (Zhao et al., 2015), and
other cities (10-15% in Tianjin, Haining, Zhongshan and Deyang; Zhou et al. (2016)).
Organic matter (OA =WSOA+WIOA($\mu g\ m^{-3}$) exhibited similar seasonal variations as
$PM_{2.5}$, and ranked in the order: winter (29.6±11.4) > fall (20.0±11.6)>spring
(17.8±3.9)>summer (12.9±1.2). The average mass fraction of OA in $PM_{2.5}$ was 20.3%
during the sampling period.



As illustrated in Fig. 8, the OC/EC ratios varied in different seasons and were
largest in winter (5.16) followed by spring (2.38), summer (1.88) and fall (1.75). The
largest OC/EC ratio occurred in winter, indicating that secondary organic carbon (SOC)
was likely a significant component of $PM_{2.5}$ in winter (Chow et al., 2005), however, the
high OC/EC ratio may be influenced by biomass burning and/or coal combustion
emissions during wintertime too. A number of previous works about the carbonaceous
aerosols in the YRD region also showed that highest OC/EC ratio occurred in winter and
the ratio was often larger than 2, such as Shanghai (6.35) (Zhao et al., 2015), Nanjing
(2.8)(Li et al., 2015), in consistent with our current results in Changzhou.

**3.5 PAHs analysis with GC-MS and SP-AMS**
The average concentrations of the 18 individual PAH and total PAHs (ΣPAHs) in
winter and spring are listed in Table 3. It can be seen that InP (% of total PAHs:
12.6-14.8%), BghiP (10.8–12.3%) and Chr (10.4–11.0%) were the three most abundant
PAHs species, followed by BbF (8.69-9.39%), BaP (7.37-8.29%), BeP (5.83-8.61) and
BaA (4.53-8.27%). The $\Sigma$PAHs in $PM_{2.5}$ were found in the range of 14.0-365.7 ng m$^{-3}$
(mean: 140.25 ng m$^{-3}$) and 8.9-91.3 ng m$^{-3}$ (mean: 41.42 ng m$^{-3}$) in winter and spring,
respectively. The ΣPAHs concentrations in this study are higher than those reported in
Zhenzhou (39 and 111 ng/m$^3$ in spring and winter)(Wang et al., 2014) and Shanghai
(13.7 ng m$^{-3}$ in spring) (Wang et al., 2015), but lower than that reported in many sites of
Liaoning Province (75-1900 ng m$^{-3}$) (Kong et al., 2010). PAHs with medium (4 rings)
and high molecular weights (5-6 rings) (MMW and HMW) accounted for the majority
of PAHs (88.9% in winter and 79.4% in spring). It is well known that MMW and HMW
PAHs are usually associated with coal combustion and vehicular emissions (Wang et al.,
2015). Prior study in Nanjing (He et al., 2014) also showed the significant contribution
of traffic exhaust to some PAHs including BbF, Chr, Flu, InP, BeP, and BghiP, which in
total accounted for more than 53% of the total PAHs.
The diagnostic ratios of selected PAHs including Phe/(Ant+Phe), BaP/BghiP,
Flua/(Flua+Pyr), BaP/(BaP+Chr) and Phe/(Ant+Phe) can be used to further distinguish
the emission sources of PAHs (Szabó et al., 2015). As suggested previously (Feng et al.,





2015;Saldarriaga-Noreña et al., 2015), traffic source was characterized with a ratio of
BaP/BghiP>0.6, and ratios of Flua/(Flua+Pyr) <0.4, 0.4-0.5, >0.5 indicate sources of
petrogenic, fossil fuel combustion and coal/wood combustion, respectively. In this work,
the Bap/BghiP of 0.61 (winter) and 0.76 (spring) and Flua/(Flua+Pyr) ratios of 0.47
(winter) and 0.50 (spring), all suggest that local vehicular/fossil fuel combustion
emissions could be a prominent contributor to particulate PAHs in Changzhou, and
contribution from long-range transport was thus minor. Meanwhile, BaP/(BaP+Chr)
ratio of 0.40 (winter ) and 0.44 (spring) also points to the source from gasoline emission
(Khalili et al., 1995). However, the Phe/(Ant+Phe) ratio of 0.89 (winter) and 0.86
(spring) indicate the coal combustion might be also an important source of PAHs.

On the other hand, by using the SP-AMS, we also identified a series of PAH ions,

i.e., $C_{16}H_{10}^+$ (*m/z* 202), $C_{17}H_{12}^+$ (*m/z* 216), $C_{18}H_{10}^+$ (*m/z* 226), $C_{18}H_{12}^+$ (*m/z* 228), $C_{19}H_{12}^+$
(*m/z* 240), $C_{19}H_{14}^+$ (*m/z* 242), $C_{20}H_{10}^+$ (*m/z* 250), $C_{20}H_{12}^+$ (*m/z* 252), $C_{21}H_{12}^+$ (*m/z* 264),
$C_{21}H_{14}^+$ (*m/z* 266), $C_{22}H_{12}^+$ (*m/z* 276), $C_{23}H_{12}^+$ (*m/z* 288), $C_{23}H_{14}^+$ (*m/z* 290), $C_{24}H_{12}^+$
(*m/z* 300), $C_{24}H_{14}^+$ (*m/z* 302), $C_{25}H_{16}^+$ (*m/z* 316), $C_{26}H_{14}^+$ (*m/z* 326), and $C_{26}H_{16}^+$ (*m/z*
328), as proposed by Dzepina et al. (2007), confirming the existence of PAHs in
ambient particles in Changzhou. Note many PAH ions identified by the SP-AMS were
not measured by the GC-MS, and the PAH compound DBA which is determined by the
GC-MS was not detected by the SP-AMS. This reflects the different sensitivities and
responses to the particle-bound PAHs of these two techniques. Table 4 shows the
correlation (*r*) coefficients of the concentrations of a few selected PAHs, and the mass
ratios of their concentrations measured by both the GC-MS and SP-AMS (results for
SP-AMS were based on measurements of all samples, while results for GC-MS were for
23 samples in winter and spring). It can be seen that the concentrations of
GC-MS-determined PAHs correlated very well with each other (*r*>0.92), while the mass
loadings determined by the SP-AMS correlated relatively weak. Also, the mass ratios
determined from these two instruments were also different. The inconsistencies may be
due to the following reasons: (1) the SP-AMS break parent PAH molecules into
fragments due to 70 ev EI, thus concentration of a specific PAH ion from the SP-AMS





cannot represent its corresponding parent PAH compound, while GC-MS determines the
concentration of the molecular PAH compound; (2) One PAH ion in the SP-AMS
HRMS may be combination of a few PAHs compounds with the same molecular
weights; (3) Sensitivities and responses to the trace amount of PAHs of the SP-AMS
may be different, thus may lead to uncertainties of the PAHs quantification.
Nevertheless, combining GC-MS and SP-AMS to improve the PAH measurements by
the SP-AMS is valuable, and will be the subject of our future work.

**3.6 Source apportionment of WSOA**
**3.6.1 WSOA mass spectral profile**
To gain further insights into the particulate OA characteristics, we performed the
SP-AMS analyses on the water extract of the $PM_{2.5}$ samples, with a focus on OA. The
averaged high resolution mass spectra (HRMS) of WSOA classified by six ion
categories and five elements are shown in Fig. 9, and the corresponding inset pie charts
represent the mass percentages of the ion families and elements, respectively. As
illustrated in Fig. 9a, the $C_xH_y^+$ ion family accounts for 38.7% of the WSOA HRMS,
followed by $C_xH_yO_2^+$ (28.0%), $C_xH_yN_p^+$ (17.7%) and $C_xH_yO^+$ (10.4%). It is worth to
mention that we found that the $C_xH_yN_p^+$ ions contributed significantly, and the organic N
(ON) could occupy 8.4% of the total WSOA mass (Fig.9 b). The average concentration
of water-soluble organic nitrogen (WSON) over the sampling period was 1.5 µg N m$^{-3}$
(114 nmol N m$^{-3}$), which is in fact lower than those measured in Beijing (226 nmol N
m$^{-3}$) (Duan et al., 2009), Qingdao (129-199 nmol N m$^{-3}$) (Shi et al., 2010), Xi,an (300
nmol N m$^{-3}$) (Ho et al., 2015). The concentration of water-soluble inorganic nitrogen
(WSIN, N from ammonium, nitrate and nitrite) was 8.7 µg N m$^{-3}$ base on Table 2, and
thus the WSON content corresponds to 14.9% of water-soluble nitrogen (WSN = WSON
+WSIN). This values is also lower than those in Beijing (~30%) (Duan et al., 2009),
Qingdao (19-22.6%), and Xi,an (22-68%) (Ho et al., 2015).
Nevertheless, the level of ON measured here are a few times higher than those
observed in other locations from AMS measurements (typically 1-3%) (Xu et al., 2014),
likely due to the following reasons: First, previous studies were online measurements on





non-refractory submicron aerosols, while it is likely that the supermicron fine particles
(1-2.5 μm) contain significant nitrogen-containing species, as observed before for
marine aerosols (Violaki and Mihalopoulos, 2010). Secondly, we measured only the
water-soluble fraction of OA, which may concentrate more nitrogen-containing species
(partially from aqueous-phase processing). Thirdly, a recent study reveals that fossil fuel
combustion-related emission can be a dominant source of ammonia in urban area, it thus
can act as a significant contributor to amines as amines are often co-emitted with
ammonia (Ge et al., 2011b); these amines can be neutralized by inorganic or organic
acids and since aminium salts are highly hygroscopic (Ge et al., 2011a), they might be
enriched in the WSOA, and generated significant $C_xH_yN_p^+$ ions. Nevertheless, more
AMS analyses on the water-extracted $PM_{2.5}$ samples collected from other locations
should be conducted to further verify the abundance of ON species in the AMS mass
spectra of WSOA.

Overall, the average elemental ratios of the WSOA are 0.36 for O/C, 1.54 for H/C,

0.11 for N/C and 1.74 for OM/OC (Fig. 9a). WSOA is on average comprised of 61.4% C
7.2 % H, 22.9% O, 8.4% N and a negligible fraction (0.2%) of S (Fig. 9b). Except for the
enrichment of ON, other results are similar with other online AMS measurement results,
such as in Fresno (Ge et al., 2012a).
**3.6.2 WSOA sources from PMF analysis**

The PMF analysis of the WSOA HRMS matrix identified four OA factors,

including two primary OA (POA) factors, named as the nitrogen-enriched
hydrocarbon-like OA (NHOA), and cooking-relevant OA (COA), and two secondary
OA factors which are a less oxidized oxygenated OA (LO-OOA) and a more oxidized
oxygenated OA (MO-OOA), as shown in Fig. 10.

The NHOA factor had a low O/C ratio (0.14), and was abundant in $C_xH_y^+$ ions

(33.8%) and the NHOA time series also varied closely with those ions, showing its
common feature as traffic OA. In particular, the factor was rich in $C_xH_yN_p^+$ ions (43.1%),
as a result, it shows a much higher N/C ratio (0.26, Fig. 10a) than other factors, and
correlated well with $CHN^+$ ($r^2$=0.82), $CH_4N^+$ ($r^2$=0.90), and $CH_2N^+$ ($r^2$=0.70), and



$C_2H_4N^+$ ($r^2$=0.76) (Fig. 10b). The N-containing ions in the NHOA MS were dominated
by the reduced ions ($C_xH_yN^+$) rather than oxidized ones ($C_xH_yO_zN^+$), suggesting that
amino compounds were likely the major ON species, and was in consistent with our
hypothesis aforementioned in Section 3.6.1 that they were mainly from traffic emissions.
Nevertheless, future studies should be conducted to investigate in details the
contribution of traffic source to the atmospheric ON species.

The COA had a low O/C ratio of 0.14 and contained mainly reduced $C_xH_y^+$ ions

(60.8%) as well, representing its primary origin. Its mass spectrum is characterized by
peaks at *m/z* 55 (significant $C_3H_3O^+$) and *m/z* 57 (significant $C_3H_5O^+$). The abundance of
$C_3H_3O^+$ at *m/z* 55 and $C_3H_5O^+$ at *m/z* 57 is a spectral feature of cooking OA, and the
overall COA MS and O/C ratios are also similar to the COA factors reported in other
studies, such as in Beijing (Sun et al., 2016). The COA time series also correlated well
with other cooking-related marker ions, such as $C_5H_8O^+$ ($r^2$=0.58), $C_6H_{10}O^+$ ($r^2$ = 0.54),
$C_7H_{12}O^+$ ($r^2$=0.45), consistent with the observations from many previous studies (e.g.,
Sun et al., 2011b;Ge et al., 2012a). All these results indicate its feature as
cooking-related OA. However, the ratio of COA/$C_6H_{10}O^+$ (622.0) in this study was
much higher than that obtained in winter in Fresno and New York City (~180), likely
due to we only detected the water-soluble fraction of COA.

The LO-OOA MS profile exhibited characteristics of oxidized OA with enhanced

signals at *m/z* 29 ($CHO^+$), *m/z* 43 (mainly $C_2H_3O^+$) and other oxygenated ions. Tight
correlations between time series of LO-OOA and $CHO^+$ ($r^2$=0.84), and $C_2H_3O^+$ ($r^2$=0.54)
were also observed. Moreover, we also noticed relatively high signals of the BBOA
tracer ions $C_2H_4O_2^+$ and $C_3H_5O_2^+$ in the LO-OOA MS, and found good correlations
between LO-OOA and BBOA tracers ($r^2$=0.76 with $C_2H_4O_2^+$, and $r^2$=0.86 with
$C_3H_5O_2^+$), indicating possible influence from biomass burning on the LO-OOA. Thus,
we compared mass fraction of LO-OOA to total OA in different seasons assuming that
LO-OOA would increase in straw-burning seasons given that it could be influenced by
BBOA. Figure S6 showed the mass fraction of four factors during straw-burning seasons
(spring, summer) and non-straw burning seasons (fall, winter). No obvious difference




for LO-OOA fraction was found, thus this factor is in fact not heavily influenced by
BBOA. Furthermore, the O/C and OM/OC ratios were 0.34 and 1.70, well within the
O/C range of less-oxidized OA factors identified in other studies (Jimenez et al., 2009),
but beyond the O/C range of typical BBOA (0.18-0.26) (He et al., 2010). On the other
hand, the MO-OOA factor had prominent peaks at $m/z$ 28 (mainly $CO^+$) and $m/z$ 44
(mainly $CO_2^+$), and was dominated by $C_xH_yO_1^+$ (36.6%) and $C_xH_yO_2^+$ ions (29.0%) (Fig.
10a). As a result, MO-OOA had a very high O/C ratio of 1.04, showing that it is heavily
aged and processed OA component. Correspondingly, its time series correlated well with
the secondary OA tracer ions, such as $CO_2^+$ ($r^2$=0.87) (Fig. 10b), $C_2H_4O^+$ ($r^2$=0.45) and
$C_2H_3O^+$ ($r^2$=0.53), etc.
The $f44$ (mass fraction of $m/z$ 44 to the total OA) versus $f43$ (mass fraction of $m/z$ 43
to the total OA, defined by Ng et al. (2010)), can be used to investigate the degree of
oxygenation of the identified factors. As presented in Fig. 11a, apart from NHOA, other
three factors (COA, LO-OOA and MO-OOA) all fall within the triangular region.
MO-OOA located at the upper position with a higher $f44$ of 0.28, while LO-OOA
located at the lower position of plot as it had a high fraction of $f43$ (0.09). This
distribution of the four factors is also consistent with other studies.
The mass contributions of the four factors to total WSOA over the whole year are
23.9% for NHOA, 31.2% for COA, 15.3% for LO-OOA and 29.7% for MO-OOA (Fig.
11b). POA (=NHOA+COA) overweighed SOA (=LO-OOA+MO-OOA) mass, showing
the dominant role of local anthropogenic emissions to the aerosol pollution in
Changzhou, similar to that observed in Nanjing (Wang et al., 2016b). However, during
spring and winter, SOA contributions dominate over POA, indicating significant SOA
formation in particular the MO-OOA during cold seasons, which is in agreement with
the OC/EC results.

**4. Conclusions**
We presented here the comprehensive characterization results on the PM$_{2.5}$ samples
collected across one year in Changzhou City, located in the YRD region of China. The



species we quantified including WSIIs, trace metals, EC, WSOA, WIOA and also PAHs,
can reproduce on average ~80% mass of the PM$_{2.5}$ (108.3 μg m$^{-3}$). WSIIs were the major
component, accounting for 52.1% PM$_{2.5}$ mass, and NO$_3^-$, SO$_4^{2-}$, NH$_4^+$ were three most
abundant ions. The organic matter (the sum of WSOA and WIOA) occupied ~20%
PM$_{2.5}$ mass, and EC accounted for ~5% PM$_{2.5}$ mass. Trace metal elements accounted for
~6% and ~2% PM$_{2.5}$ mass during winter and spring. Total PAHs concentrations were
found to be at a relatively high concentration of 140.25 ng m$^{-3}$ in winter, above three
times the average mass loading of 41.42 ng m$^{-3}$ in spring, both with InP, BghiP and Chr
as the three most abundant PAHs. Average mass ratio of NO$_3^-$/SO$_4^{2-}$ was 1.21,
suggesting a significant role of traffic emissions, which is in consistent with the source
analyses results based on the diagnostic ratios of the selected PAHs (BaP/BghiP,
Flua/(Flua+Pyr) and BaP/(BaP+Chr)). In addition, a high Cl$^-$/Na$^+$ ratio and the
diagnostic ratio of Phe/(Ant+Phe) indicated also the contribution from coal combustion,
in particular during winter.
In order to obtain further information regarding particle source, we analyzed the
WSOA using SP-AMS and conducted PMF analyses on the HRMS of WSOA. Four OA
factors including NHOA, COA, LO-OOA and MO-OOA were identified. The mean
mass contribution of POA (=NHOA+COA) was larger than that of SOA
(=LO-OOA+MO-OOA), revealing that local anthropogenic activities are the major
drivers of PM pollution in Changzhou. Nevertheless, during cold seasons, SOA mass
contribution increased, indicating significant role of secondarily formed species as well,
thus reduction of air pollution in Changzhou should be paid on the strict emission
control of both primary particles and the gaseous secondary aerosol precursors. One
interesting finding in this work is the enrichment of organic nitrogen species in WSOA,
and source analysis indicates that traffic emissions can be a significant contributor to
these species, which warrants more detailed investigations in the future. Also, more
offline samples should be collected to achieve a more robust PMF analyses.
Simultaneous online AMS measurement on the fine particles and measurements of
gaseous species (SO$_2$, NO$_2$, O$_3$, CO and some volatile organic compounds) are also



essential to better understand the aerosol characteristics, and to implement proper
measures to abate the air pollution in this region.

**Acknowledgements**
This work was supported the Natural Science Foundation of China (Grant Nos.
(21407079 and 91544220), the Jiangsu National Science Foundation (BK20150042),
Specially-Appointed Professors Foundation (for X.G.), the Major Research
Development Program of Jiangsu Province (BE2016657 and BY2016030-15), the open
fund by Jiangsu Key Laboratory of Atmospheric Environment Monitoring and Pollution
Control (KHK1409),. We would also like to acknowledge Mr. Gang Li from Chinese
academy of Science for providing us the OC/EC measurements.

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



Table 1. Average meteorological parameters during four seasons

| Parameters | Spring | Summer | Fall | Winter |
|---|---|---|---|---|
| RH (%) | 57.3±11.4 | 61.1±11.8 | 65.5±10.9 | 62.3±10.6 |
| T($^{o}$C) | 13.1±4.0 | 32.1±4.3 | 21.6±2.3 | 5.6±1.8 |
| WS(m s$^{-1}$) | 1.1±0.4 | 1.6±0.6 | 0.9±0.4 | 0.8±0.3 |
| WD[a] | SE | E,W,SE | E | W |

[a] Refer to prevailing wind directions, E—East, SE—Southeast, W—West.





Table 2. Summary of the mean concentrations (with one standard deviation) for the PM$_{2.5}$ and all
quantified components in four seasons and the whole sampling period.

| Species (µg m$^{-3}$) | Spring | Summer | Fall | Winter | Annual |
|---|---|---|---|---|---|
| **PM$_{2.5}$** | **106.0±24.4** | **80.9±37.7** | **103.3±28.2** | **126.9±50.4** | **108.3±40.8** |
| **WSIIs** | **66.5±17.2** | **35.0±20.2** | **51.0±17.2** | **66.8±23.6** | **56.4±22.9** |
| Sulfate | 17.3±4.8 | 15.8±9.8 | 17.2±6.2 | 18.7±7.6 | 17.5±7.1 |
| Nitrate | 26.4±8.7 | 6.8±6.2 | 17.0±9.0 | 24.1±11.8 | 19.3±11.6 |
| Ammonium | 14.8±4.2 | 8.2±4.3 | 11.2±3.2 | 13.1±3.7 | 12.0±4.2 |
| Other ions | 8.0±2.3 | 4.2±2.9 | 5.6±1.5 | 10.9±3.4 | 7.6±3.7 |
| % of PM$_{2.5}$ | 62.6±4.9 | 41.1±7.4 | 49.0±8.5 | 50.4±7.3 | 52.1±9.7 |
| **TC** | **16.0±3.3** | **12.1±1.6** | **21.0±11.8** | **22.3±8.6** | **19.2±9.3** |
| OC | 11.2±2.6 | 7.9±0.8 | 13.2±7.8 | 18.3±8.1 | 13.8±7.5 |
| EC | 4.8±0.9 | 4.2±1.2 | 7.7±4.5 | 4.0±0.9 | 5.4±3.2 |
| % of PM$_{2.5}$ | 15.3±2.5 | 17.5±6.5 | 19.7±8.2 | 20.1±3.3 | 18.1±6.1 |
| **OA** | **17.8±3.9** | **12.9±1.2** | **20.0±11.6** | **29.6±11.4** | **21.8±11.3** |
| WSOA | 13.1±2.8 | 11.0±2.2 | 14.1±6.5 | 23.4±8.0 | 16.7±7.9 |
| WIOA | 4.8±2.6 | 1.9±1.8 | 5.9±7.2 | 6.1±10.6 | 5.2±7.6 |
| % of PM$_{2.5}$ | 17.1±3.0 | 19.0±7.8 | 18.7±8.1 | 23.9±5.5 | 20.1±7.0 |
| **PAHs (ng m$^{-3}$)** | **41.42±24.7** | | | **140.25±60.2** | |
| **Trace elements** | | | **2.77±1.15** | **6.38±3.14** | |
| **OA+EC+WSIIs** | **89.1±20.9** | **52.2±21.5** | **81.5±29.6\*** | **106.8±35.9\*** | **83.7±32.1** |
| **% of PM$_{2.5}$** | **84.2±5.5** | **65.9±4.8** | **78.9±14.9\*** | **84.2±11.7\*** | **77.3±11.6** |

*These values also include contributions from trace elements.











Table 3. Mean concentration (ng m$^{-3}$) and mass fractions (%) of individual PAH to the total
PAHs.

| PAH compounds | Number of rings | Molecular formula and molecular weight (MW) | **Winter** | | **Spring** | |
|---|---|---|---|---|---|---|
| | | | Conc. (ng m$^{-3}$) | % of total | Conc. (ng m$^{-3}$) | % of total |
| NaP | 2-rings | $C_{10}H_8$,128 | 10.12 | 7.22 | 2.60 | 6.28 |
| Acy | | $C_{12}H_8$,152 | 0.16 | 0.12 | 0.08 | 0.20 |
| Ace | | $C_{12}H_{10}$,154 | 0.15 | 0.11 | 0.34 | 0.83 |
| Flu | 3-rings | $C_{13}H_{10}$,166 | 1.19 | 0.85 | 1.70 | 4.11 |
| Phe | | $C_{14}H_{10}$,178 | 3.54 | 2.52 | 3.24 | 7.83 |
| Ant | | $C_{14}H_{10}$,178 | 0.46 | 0.33 | 0.54 | 1.31 |
| Flua | | $C_{16}H_{10}$,202 | 8.05 | 5.74 | 2.57 | 6.21 |
| Pyr | | $C_{16}H_{10}$,202 | 8.93 | 6.37 | 2.43 | 5.87 |
| BaA | 4-rings | $C_{18}H_{12}$, 228 | 11.6 | 8.27 | 1.88 | 4.53 |
| Chr | | $C_{18}H_{12}$, 228 | 15.41 | 11.0 | 4.32 | 10.43 |
| BbF+BjF | | $C_{20}H_{12}$, 252 | 12.19 | 8.69 | 3.89 | 9.39 |
| BkF | | $C_{20}H_{12}$, 252 | 5.58 | 3.98 | 1.87 | 4.50 |
| BaP | 5-rings | $C_{20}H_{12}$, 252 | 10.33 | 7.37 | 3.43 | 8.29 |
| BeP | | $C_{20}H_{12}$, 252 | 12.08 | 8.61 | 2.42 | 5.83 |
| DBA | | $C_{22}H_{14}$, 278 | 2.53 | 1.8 | 0.42 | 1.02 |
| InP | 6-rings | $C_{22}H_{12}$, 276 | 20.74 | 14.8 | 5.23 | 12.62 |
| BghiP | | $C_{22}H_{12}$, 276 | 17.18 | 12.3 | 4.46 | 10.76 |
| LMW-PAHs | 2-3 rings | | 15.62 | 11.1 | 8.50 | 20.6 |
| MMW-PAHs | 4-rings | | 43.99 | 31.4 | 11.20 | 27.0 |
| HMW-PAHs | 5-6 rings | | 80.63 | 57.5 | 21.72 | 52.4 |
| ΣPAHs | | | 140.25 | 100.0 | 41.42 | 100.0 |




Table 4 Cross-correlation coefficients ($r$) of the measured concentrations of the PAH speciesand
ratios of the mean concentrations between these species from GC-MS (bold) and SP-AMS
(itlaic).

| PAHs | $C_{16}H_{10}$ | $C_{18}H_{12}$ | $C_{20}H_{12}$ | $C_{22}H_{12}$ | Ratio (GC) | Ratio (SP-AMS) |
|---|---|---|---|---|---|---|
| $C_{16}H_{10}$ | 1 | *-0.250* | *-0.062* | *-0.140* | **$C_{16}H_{10}/C_{16}H_{10}$=1** | *$C_{16}H_{10}^{+}/C_{16}H_{10}^{+}$=1* |
| $C_{18}H_{12}$ | **0.952** | 1 | *0.572* | *0.528* | **$C_{16}H_{10}/C_{18}H_{12}$=0.84** | *$C_{16}H_{10}^{+}/C_{18}H_{12}^{+}$=0.43* |
| $C_{20}H_{12}$ | **0.936** | **0.994** | 1 | *0.771* | **$C_{16}H_{10}/C_{20}H_{12}$=0.36** | *$C_{16}H_{10}^{+}/C_{20}H_{12}^{+}$=0.56* |
| $C_{22}H_{12}$ | **0.925** | **0.986** | **0.993** | 1 | **$C_{16}H_{10}/C_{22}H_{12}$=0.35** | *$C_{16}H_{10}^{+}/C_{22}H_{12}^{+}$=1.17* |

$C_{16}H_{10}$: Flua+Pyr; $C_{18}H_{10}$: BaA+Chr; $C_{20}H_{12}$: BbF+BjF+BkF+BaP+BeP;
$C_{22}H_{12}$: BghiP+InP+DBA





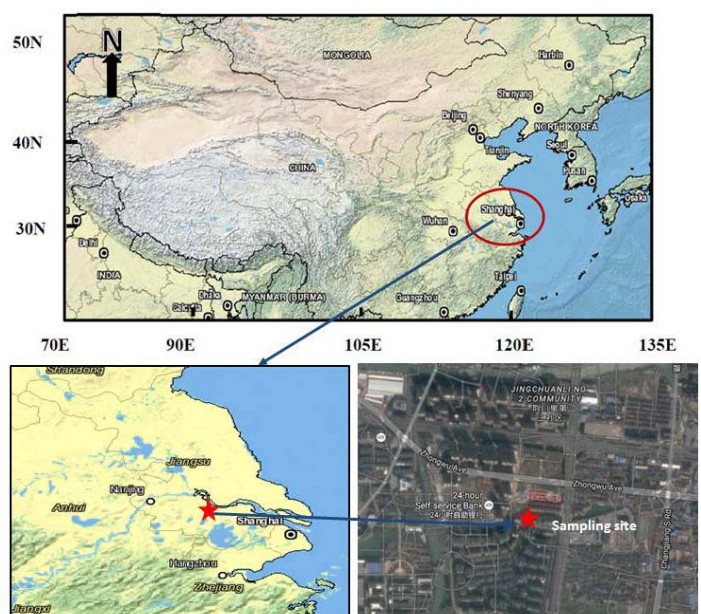


Figure 1. Schematic map of the sampling site, its surroundings and location.





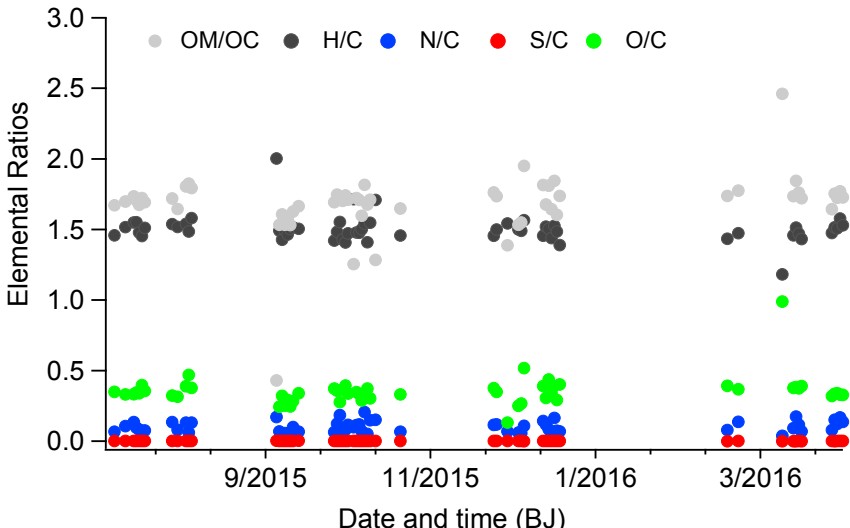


Figure 2. The atomic elemental ratios for the water-soluble organic matter (WSOA) determined
by the SP-AMS.









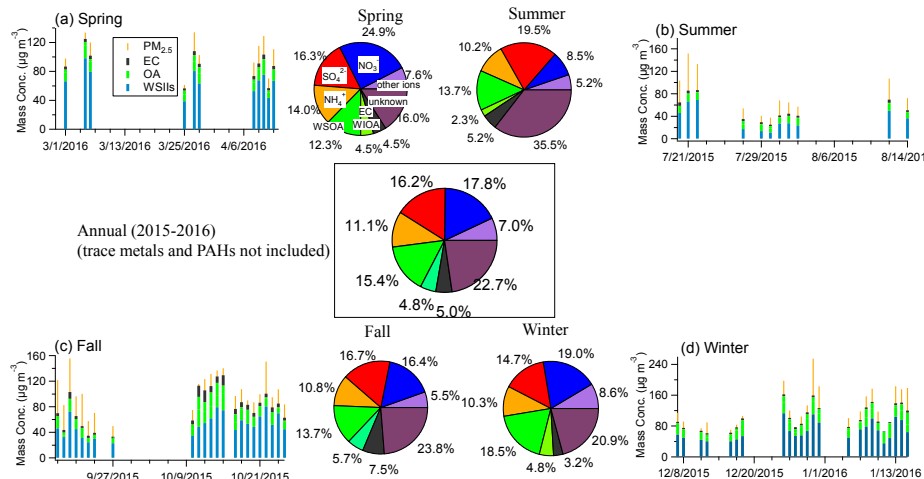



Figure 3. Reconstructed mass (=OA+EC+WSIIs) vs. PM$_{2.5}$ mass from gravimetric measurement
(PM$_{2.5}$) in (a) sping, (b) summer, (c) fall, (d) winter, and annual. Corresponding pie charts show
the mass percentages of different species to the PM$_{2.5}$ mass (trace elements and PAHs are not
included due to sample limitations).



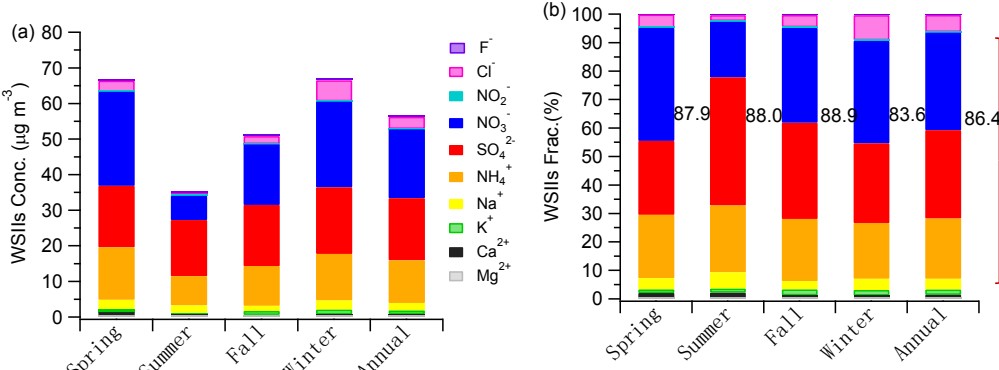



Figure 4. (a) Seasonal variations of average mass concentrations and (b) mass fractional

contributions of WSIIs in PM$_{2.5}$ in Changzhou during 2015-2016. The values marked in (b) are

the fractions of three major ions (NO$_3^-$+SO$_4^{2-}$+NH$_4^+$) to the total WSIIs.






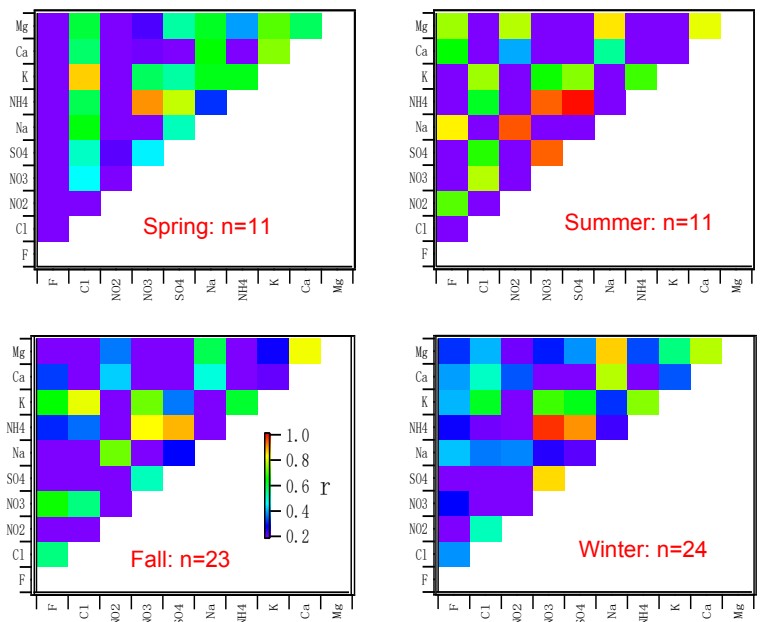


Figure 5. Image plots showing the cross correlation coefficients between water-soluble ions in
PM$_{2.5}$ in four seasons. Boxes are colored by correlations (*r*).





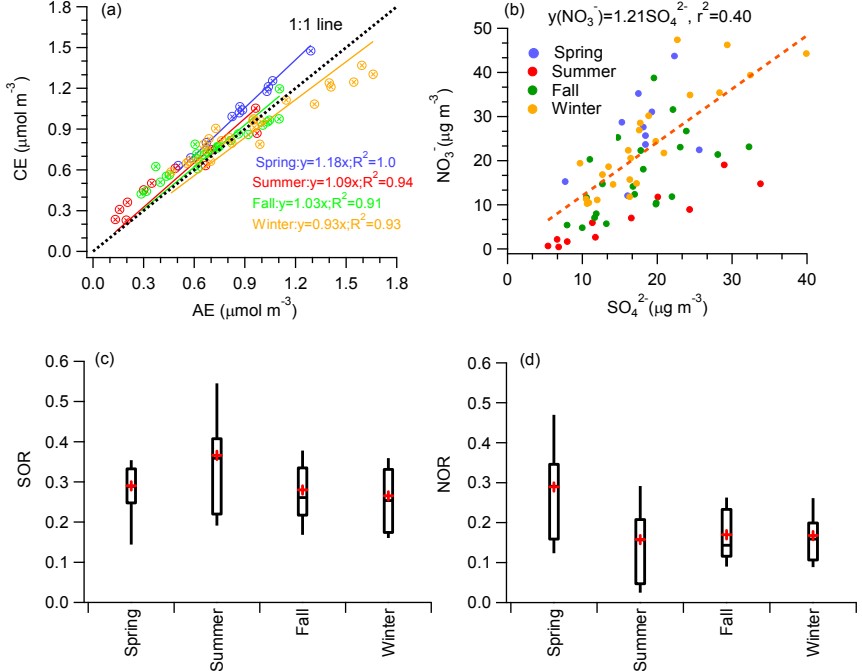

Figure 6. (a) Scatter plots of molar concentrations of cations vs. anions, (b) scatter plots of $NO_3^-$ vs. $SO_4^{2-}$ concentrations, (c-d) SOR and NOR value during four seasons. In (a), the dashed line refers to 1:1 line. In (b), the dashed line was the averaged fitted line, representing $NO_3^-/SO_4^{2-}$ ratio during the entire period. Data in different season are shown by different colors for comparison. Linear regression equations were also presented. In (c-d), the crosses represent the mean, the middle bars represent the median, the top and bottom of the box represents the 75th and 25th percentile, respectively, and the top and bottom whiskers represent the 90th and 10th percentile, respectively.



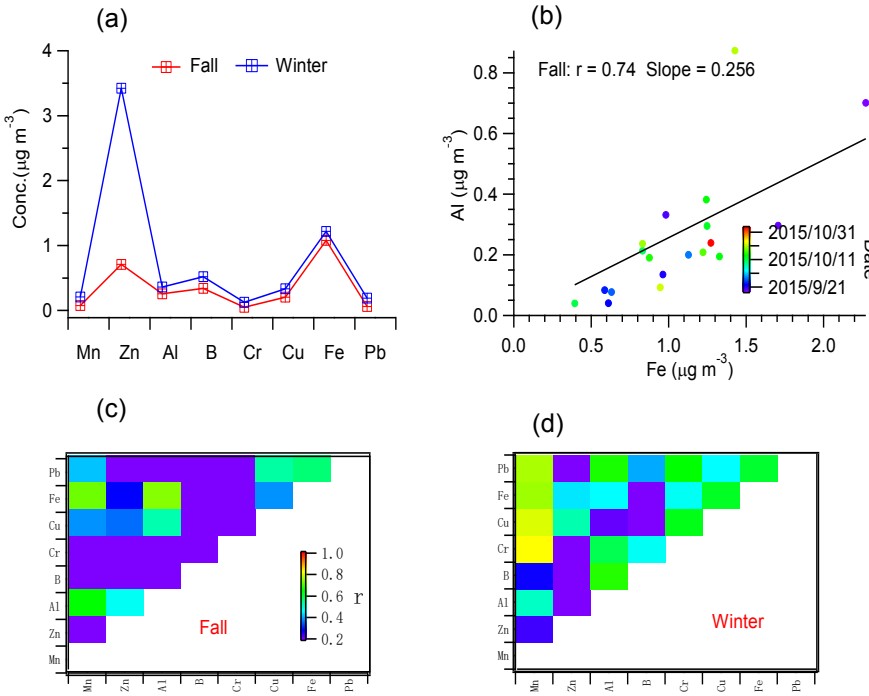


Figure 7. (a) Mean mass concentrations of trace elements determined for fall and winter, (b)
Scatter plots of Al and Fe in fall, and (c-d) cross-correlation coefficients (*r*) among different
trace elements in fall and winter, respectively.







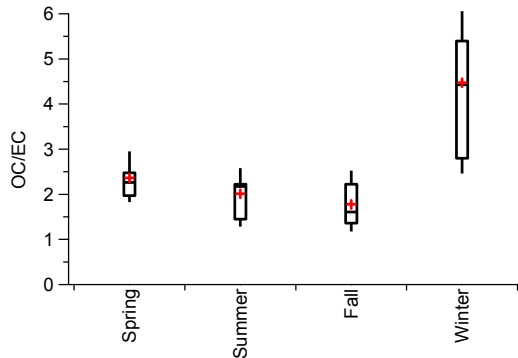


Figure 8. Average OC/EC ratios measured in four seasons (symbols of the box plots are the
same as described in Figure 6.)



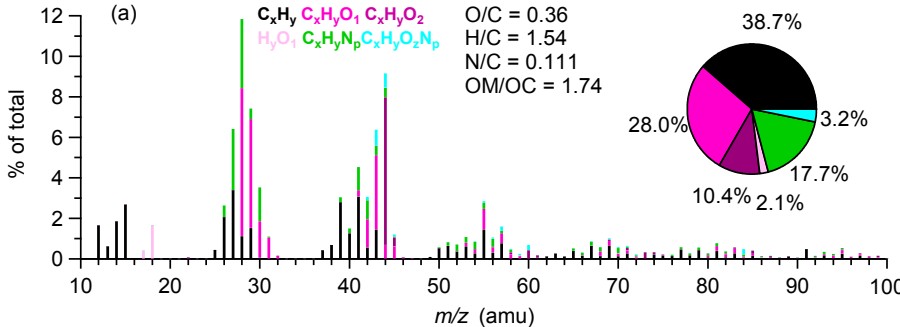

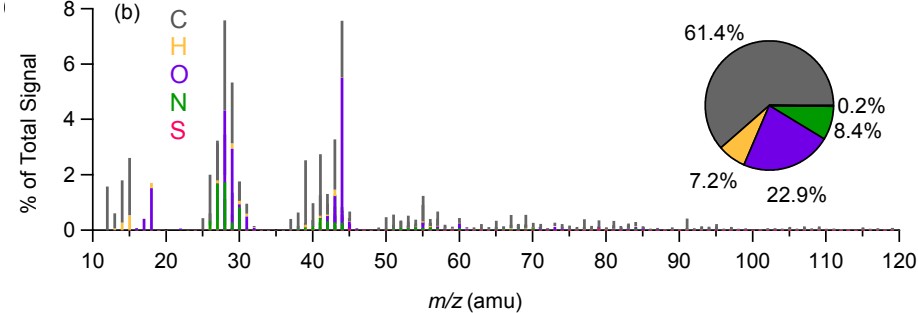


Figure 9. (a) High-resolution mass spectral profile of the WSOA measured by the SP-AMS
(Mass spectrum is classified and colored by six ion families; pie chart shows the mass
contributions of each ion family to the total MS), (b) Average mass spectrum classified by five
elements (C, H, O, N, and S) (inset pie chart shows mass contributions of the five elements,
respectively).





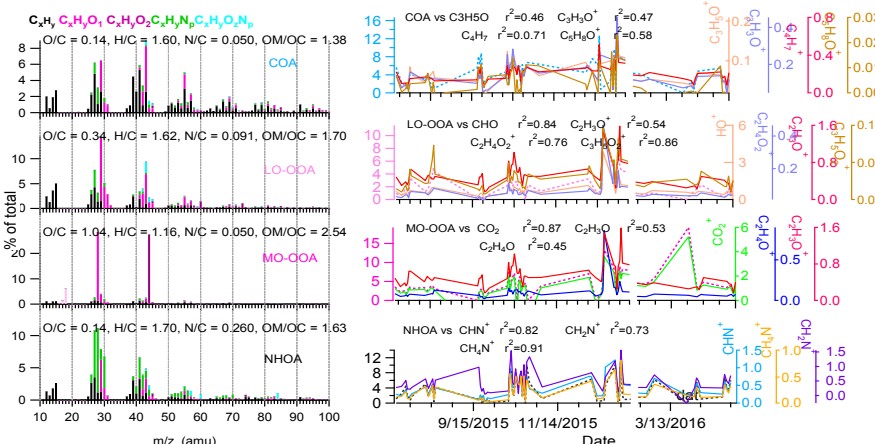

Figure 10. (a) High-resolution mass spectra of nitrogen-enriched hydrocarbon-like OA (NHOA),

cooking-related OA (COA), less-oxidized OA (LO-OOA) and more-oxidized OA (MO-OOA)

separated by the PMF analyses, colored by six ion categories, (b) time series of the four WSOA

factors, and corresponding tracer ions.



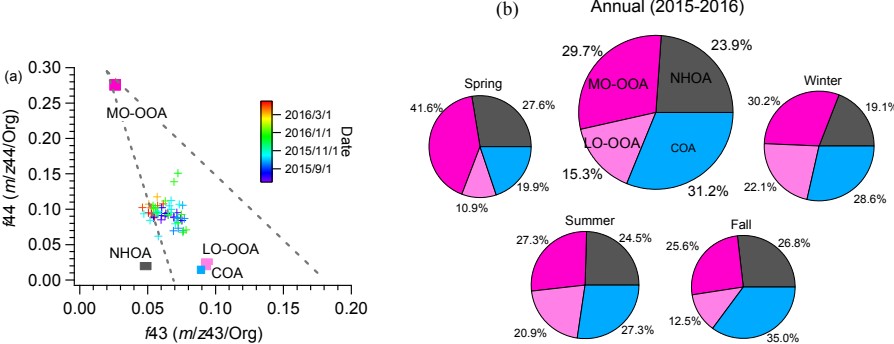


Figure 11. (a) Triangle plot of *f*44 *vs.* *f*43 for all WSOA, and the four WSOA factors identified
by the PMF analyses, (b) pie charts of the mass contributions of four WSOA factors to the total
WSOA in four seasons and the whole sampling period.