# Peer review of "Chemical characterization of fine particulate matter in Changzhou, China 1 and source apportionment with offline aerosol mass spectrometry 2 3 Zhaolian Ye1,2, Jiashu Liu1, Aijun Gu1, Feifei Feng1, Yuhai Liu1, Chenglu Bi1, Jianzhong"

_Atmospheric Chemistry and Physics, 2016_

## Referee Comment (RC1) · Anonymous Referee #2 · 3 Dec 2016

**Referee comment**

**Chemical characterization of fine particular matter in Changzhou, China and source apportionment with offline aerosol mass spectrometry**

Z. Ye et al., Atmos. Chem. Phys. Discuss., doi:10.5194/acp-2016-883, 2016

**Anonymous referee #2**

**General comments**

This manuscript reports results obtained during a long-term measurement campaign performed at Changzhou, China. The authors sampled $PM_{2.5}$ particles on filters during one year (one month per season) and used a wide range of off-line analytical techniques to determine the concentration and chemical composition of these samples.

This is a long and important effort in terms of sampling, off-line analysis and data treatment. Results reported in this manuscript will be of interest for the readers of *Atmospheric Chemistry and Physics*. I recommend its final publication after the authors address the following comments.

**Specific comments**

1) The main issue of this manuscript is the absence of discussion on the uncertainty of the results. Given that the authors used a large set of analytical techniques, they should present their uncertainties in their respective sub-section under 2.2 "Chemical analysis". This is particularly important for a few parameters which are calculated using results from two instruments, such as the concentration of water soluble organic aerosols (WSOA), which is obtained from the TOC analyzer and the OM/OC ratio of the SP-AMS.

2) Section 2.1 "Sampling site and $PM_{2.5}$ collection": the authors need to mention here the artifacts related to the filter samplings, in particular the evaporation of semi-volatile compounds during the sampling. This is particularly important for some results presented later, such as the $NO_3^-/SO_4^{2-}$ ratio. Indeed, if these species are present under the form of ammonium nitrate and ammonium sulfate (as shown in section 3.2), ammonium nitrate will evaporate faster than ammonium sulfate during the sampling. Therefore, the concentrations of nitrate correspond to a lower limit, the real concentrations should be higher, and the real $NO_3^-/SO_4^{2-}$ ratios should also be higher. Another artifact concerns the adsorption of gases, such as volatile organic compounds (VOCs), onto the sampling media and collected particles, which can have an impact on the concentration of particle-bound polycyclic aromatic hydrocarbons (PAHs).

3) Section 2.1 "Sampling site and $PM_{2.5}$ collection": according to the wind rose plots presented in Fig. S1, the sampling site was under the influence of different air masses, depending on the season. This important information is not discussed in the section 3 "Results and discussion" and the corresponding sub-sections. Did the authors perform a back trajectory analysis to check where the air masses come from during each sampling period?

4) Section 2.2.5 "Offline SP-AMS analysis": it would be interesting if the authors explain here the advantages to use the SP-AMS for off-line analysis of filter samples. This kind of analysis presents several problems compared to on-line measurements: a) it has a much lower time resolution (20 hours in this study, instead of a few minutes), b) the total concentrations and size distributions of the species cannot be directly measured (given that the water extracts must be atomized), and c) it introduces artifacts related to the filter sampling. So what are the advantages of using that instrument for off-line analysis?

5) Section 2.4 "Source apportionment of WSOA": the authors should say a few words on the robustness of this PMF analysis, given that the dataset contains only 69 samples (67 if the authors discarded two outliers) and corresponds to 20-hours averaged samples.

6) Lines 369-372: in addition to cations not measured by ion chromatography, a $NH_4^+$ measured/$NH_4^+$ predicted of 0.75 in winter can also simply be due to the presence of acids.

7) Lines 380-383: in summer, the high temperature may lead to a faster evaporation (not dissociation) of nitrate during the filter sampling. This may explain the lower $NO_3^-/SO_4^{2-}$ ratio in summer.

8) Lines 389-391: the authors mention that the sulfur oxidation ratio was higher in summer. However, according to Fig. 6c, the difference with the other seasons does not seem significant.

9) Lines 538-542: it is surprising to notice that the O/C ratio of organics remained almost constant during the four seasons (Fig. 2), while we could expect higher values in summer due to increased photochemical activities. Can the authors say a few words on this in the manuscript? Among all the results presented in this manuscript (OC/EC ratio, etc.), only a higher sulfur oxidation ratio in summer seems to show increased photochemical activities during that period.

**Technical comments**

10) Several correlation coefficients are reported throughout the manuscript. Sometimes, the authors use the Pearson's coefficient r, and sometimes the $r^2$. It will be better to be consistent and use systematically the same correlation coefficient, either r or $r^2$.

11) When at least two references are given in parentheses, please add a space after the semicolons.

12) Title (also in the supplementary material): "Chemical characterization of fine  particulate matter".

13) Line 24: "the fine  particulate matter ($PM_{2.5}$) samples".

14) Line 103: "short-term" is quite vague here. The typical duration of field campaigns with the AMS is approximately one month.

15) Lines 183 and 446: the authors may mention in the title of these two sub-sections that they are talking about particle-bound PAHs, not about gas-phase PAHs.

16) Line 276: by which factor were ions with S/N ratios between 0.2 and 2 downweighed?

17) Line 301: "Previous studies  showed that low".

18) Line 363: actually, the $NH_4^+$ measured/$NH_4^+$ predicted ratio was first presented by Zhang et al. (2007), and used in tens of papers afterwards, Young et al. (2016) being one of them. Therefore, I would suggest to replace this reference.

19) Lines 649-650: "and Pollution Control (KHK1409)~. We would".

20) Figure 7: it would be important to include error bars corresponding to the standard deviations. This is particularly important for the Zn concentration in winter: is this high value due to 1-2 outliers, or do all the samples have a high value?

21) Figure 9: please scale the x-axes of the two panels the same way (either *m/z* 10-100 or 10-120).

**References**

Young, D. E., Kim, H., Parworth, C., Zhou, S., Zhang, X., Cappa, C. D., Seco, R., Kim, S., and Zhang, Q.: Influences of emission sources and meteorology on aerosol chemistry in a polluted urban environment: results from DISCOVER-AQ California, Atmos. Chem. Phys., 16, 5427-5451, 10.5194/acp-16-5427-2016, 2016.

Zhang, Q., Jimenez, J. L., Worsnop, D. R., and Canagaratna, M.: A case study of urban particle acidity and its influence on secondary organic aerosol, Environ. Sci. Technol., 41, 3213-3219, 10.1021/es061812j, 2007.

---

## Referee Comment (RC2) · Anonymous Referee #1 · 24 Dec 2016

The manuscript by Ye et al. had a comprehensive analysis of fine particle composition in a city in Yangtze River Delta using a suite of analytical instruments including a SP-AMS. This is also one of a few studies in China to do offline analysis of aerosol particles using aerosol mass spectrometry. The data and results are clearly important for a better understanding the sources, seasonal variations, and formation mechanisms of aerosol particles in this polluted region. I recommend it for publication after addressing the following comments.

The authors assume $CO_2^+ = CO^+$ in calculating elemental ratios because of the influences of inorganic carbonate. The authors can have a better evaluation of the relationship between $CO^+$ and $CO_2^+$ by showing a scatter plot. The reason is 1) the O/C and OM/OC ratios in Figure 2 are relatively close among different seasons, which is not expected as usual; 2) the mass closure analysis in Figure 3 showed a substantially unidentified fraction, particularly in summer. For example, organic aerosol only accounts for 16% of $PM_{2.5}$ in summer. Is it due to the low OM/OC ratio? In addition, Canagaratna et al. (2015) recommended a new calibration factor for O/C, which can also increase the OM/OC ratio, and hence the total mass of organic aerosol.

Interpretation of the COA factor need to be cautious. It appears to me that defining this factor as COA is not appropriate although the spectrum has some similarities to the standard spectra of cooking aerosols. One of the reasons is the large contribution of COA in water-soluble organic aerosol (annual average 31.2%), which is much higher than those previously observed in urban cities. Another reason is the extraction efficiency of COA is typically much lower than secondary organic aerosol (Huang et al., 2014).

Check Figure 9.The spectral patterns of ion families and elements should be identical.

Correct "Particular" in the title and abstract.

Some statements need to be clarified. For example, line 33 – 34, higher nitrate than sulfate does not necessarily indicate traffic emissions although I know the authors want to say that traffic emissions are more important than stationary sources. Also, rephrase the statement in line 111 – 114.

Some linear fittings seem not appropriate to force the intercept to be zero, e.g., Figure 6b (winter) and Figure 7b.

**References:**

Canagaratna, M. R., Jimenez, J. L., Kroll, J. H., Chen, Q., Kessler, S. H., Massoli, P., Hildebrandt Ruiz, L., Fortner, E., Williams, L. R., Wilson, K. R., Surratt, J. D.,

Donahue, N. M., Jayne, J. T., and Worsnop, D. R.: Elemental ratio measurements of organic compounds using aerosol mass spectrometry: characterization, improved calibration, and implications, Atmos. Chem. Phys., 15, 253-272, 10.5194/acp-15-253-2015, 2015.

Huang, R.-J., Zhang, Y., Bozzetti, C., Ho, K.-F., Cao, J.-J., Han, Y., Daellenbach, K. R., Slowik, J. G., Platt, S. M., Canonaco, F., Zotter, P., Wolf, R., Pieber, S. M., Bruns, E. A., Crippa, M., Ciarelli, G., Piazzalunga, A., Schwikowski, M., Abbaszade, G., Schnelle-Kreis, J., Zimmermann, R., An, Z., Szidat, S., Baltensperger, U., Haddad, I. E., and Prevot, A. S. H.: High secondary aerosol contribution to particulate pollution during haze events in China, Nature, 514, 218 - 222, 10.1038/nature13774, 2014.

---

## Author Comment (AC1) · 23 Jan 2017

**Response to Reviewer's Comments**

**Manuscript Number**:   acp-2016-883

**Authors:** Zhaolian Ye, Jiashu Liu, Aijun Gu, Feifei Feng, Yuhai Liu, Chenglu Bi, Jianzhong Xu, Ling Li, Hui Chen, Yanfang Chen, Liang Dai, Quanfa Zhou, Xinlei Ge

**Response to Reviewer #1**

**General comment:** This manuscript reports the measurement results of submicron aerosols by the SPAMS in Nanjing. Recently the Aerodyne AMS has been widely used around the world, and this work presents for the first time the results using the SP-AMS in the YRD region. This is overall a very well written paper with quite thorough analyses of the data, the figures are informative and the results provide new insights regarding the aerosol chemistry in this region.

**Authors' reply:** We thank the reviewer for his positive comment, and our point-to-point replies to the reviewer's comments are listed below.

The authors assume $CO_2^+$ = $CO^+$ in calculating elemental ratios because of the influences of inorganic carbonate. The authors can have a better evaluation of the relationship between $CO^+$ and $CO_2^+$ by showing a scatter plot. The reason is 1) the O/C and OM/OC ratios in Figure 2 are relatively close among different seasons, which is not expected as usual; 2) the mass closure analysis in Figure 3 showed a substantially unidentified fraction, particularly in summer. For example, organic aerosol only accounts for 16% of PM2.5 in summer. Is it due to the low OM/OC ratio? In addition, Canagaratna et al. (2015) recommended a new calibration factor for O/C, which can also increase the OM/OC ratio, and hence the total mass of organic aerosol.

**Authors' reply:** Thanks for the suggestion. First, as recommended, we have tried to make a scatter plot of original $CO_2^+$ vs. CO+. The plot has a good correlation coefficient of $r^2$ = 0.92, but is with an abnormally high slope ~2.24. Although a previous study that also used argon as carrier gas for atomization of PM1 samples collected in Europe sites (Bozzetti et al., 2017), also found that the signal of $CO_2^+$ appeared to be systematically higher than that of $CO^+$, but the factor is much less than 2.24. Considering the different size cuts of that study and this work, it is indeed very likely that a significant portion of $CO_2^+$ could be due to carbonate, which is also in

some extent verified by the anion deficiency in Figure 6a. Thus, we believe it is likely reasonable to assume $CO_2^+$ equal to $CO^+$, as this ratio is proposed by Aiken et al. (2008), and was widely used and accepted by the AMS community. These discussion are now included in the modified manuscript.

Secondly, compared with the online AMS results, the mass fraction of organic matter was somehow low (~20%). But in fact, this OM fraction is well within the typical range of other mass closure results on $PM_{2.5}$ filter samples, as summarized in Liang et al. (2017). Possible reasons that it is lower than those online results include that online measurement cannot measure some crustal elements and also the nitrate/sulfate in supermicronmeter range.

At last, indeed Canagaratna et al. (2015) has proposed new calibration factors for O/C and H/C ratios, which can increase the OM/OC ratio, and hence the total mass of organic aerosol. In the revised manuscript, we now used the new set of calibration factors, and updated all relevant figures (Fig. 2, Fig. 3 and Fig. S4 in the supplement), tables (Table 4, which is original Table 3) and texts in the manuscript. Indeed, the OM mass fraction increased and the elemental ratios of various PMF factors. Please check the details in the revised manuscript.

Interpretation of the COA factor needs to be cautious. It appears to me that defining this factor as COA is not appropriate although the spectrum has some similarities to the standard spectra of cooking aerosols. One of the reasons is the large contribution of COA in water-soluble organic aerosol (annual average 31.2%), which is much higher than those previously observed in urban cities. Another reason is the extraction efficiency of COA is typically much lower than secondary organic aerosol (Huang et al., 2014).

**Authors' reply:** We generally agree with the reviewer. For the offline AMS-PMF analyses, due to low time resolution, the factor cannot be justified by investigating its diurnal pattern. The factor was defined as COA mainly due to its low O/C ratios and some similarities with previously reported COA mass spectra. As pointed out by the reviewer, the very high mass fraction of this factor suggests that this factor may include significant contributions from other sources in addition to cooking (we cannot conclude there is no cooking contribution as well). Also, we only analyzed the water-soluble fraction of OA, and COA was in fact has a relatively low extraction、 recovery ratio in water, as shown by Huang et al., 2014, Bozzetti et al., 2017 and Xu

et al., 2017, it is indeed not appropriate to assign this factor to COA only. We think it is reasonable to define this factor as a local POA factor, which is a mixture of contributions from anthropogenic sources cooking, coal combustion, industry, etc. Relevant figures and discussions regarding this factor were now updated in the revised manuscript.

3. Check Figure 9.The spectral patterns of ion families and elements should be identical.

**Authors' reply:** Thanks for the comment. We have checked Fig. 9 (a) and (b) to make them consistent. We also checked WSON because it was calculated from N fraction in WSOA in Fig.9 (b). The average concentration of water-soluble organic nitrogen (WSON) over the sampling period was now 1.16 $\mu$g N m$^{-3}$ (83 nmol N m$^{-3}$), replacing the original value of 1.5$\mu$g N m$^{-3}$(114 nmol N m$^{-3}$).

Correct "Particular" in the title and abstract.

**Authors' reply:** Corrected.

Some statements need to be clarified. For example, line 33 – 34, higher nitrate than sulfate does not necessarily indicate traffic emissions although I know the authors want to say that traffic emissions are more important than stationary sources. Also, rephrase the statement in line 111 – 114.

**Authors' reply:** We have rephrased the corresponding descriptions.

Some linear fittings seem not appropriate to force the intercept to be zero, e.g., Figure 6b (winter) and Figure 7b.

**Authors' reply:** We have now re-plotted the relevant figures without forcing the intercept to be zero, and also modified relevant discussions. For figure 6b, as we wanted to calculate the overall mass ratio for $NO_3^-$ vs. $SO_4^{2-}$, so we kept the current fitting by forcing the intercept to be zero.

**Response to Reviewer #2**

General comments: This manuscript reports results obtained during a long-term measurement campaign performed at Changzhou, China. The authors sampled PM2.5 particles on filters during one year (one month per season) and used a wide range of off-line analytical techniques to determine the concentration and chemical composition of these samples. This is a long and important effort in terms of sampling, off-line analysis and data treatment. Results reported in this manuscript will be of interest for the readers of Atmospheric Chemistry and Physics. I recommend its final publication after the authors address the following comments.

**Authors' reply:** We thank the reviewer for his positive comment, and our point-to-point replies to the reviewer's comments are listed below.

1) The main issue of this manuscript is the absence of discussion on the uncertainty of the results. Given that the authors used a large set of analytical techniques, they should present their uncertainties in their respective sun-section under 2.2 "Chemical analysis". This is particularly important for a few parameters which are calculated using results from two instruments, such as the concentration of water soluble organic aerosols (WSOA), which is obtained from the TOC analyzer and the OM/OC ratio of the SP-AMS.

**Authors' reply:** Thanks for the suggestion. Now in Section 2.2, we added a new Table 2 which lists the uncertainties and the detection limits of different analytical techniques used in this study. The measurement uncertainties were typically calculated as 3 times the standard deviation on replicate measurements on blank filters. Note the uncertainty of the OM/OC ratio is 6%, which is reported by Aiken et al. (2008). Also, since the WSOA concentrations were based on two instruments (SP-AMS) and TOC, so the uncertainty of WSOA was calculated as the sum of squares of the uncertainties of OM/OC and WSOC analyses.

Table 2 Summary of species, analytical instruments, uncertainties and detection limits.

| Species/Parameters | Analytical instruments | uncertainty | Detection limits |
|---|---|---|---|
| Water soluble ions | Ion chromatography | 3.5-7.0% | 3-20 µg L$^{-1}$ |
| Trace elements | ICP-OES | 10.3-18.5% | 16.3% |
| OC, EC | Thermal-Optical Carbon Anlyzer | <12% | 30-80 ng m$^{-3}$ for OC and 30 ng m$^{-3}$ for EC (Mirante et al., 2014) |
| WSOC | TOC analyzer | 3.4-6.0% | 5.0 µg L$^{-1}$ |
| PAH | GC-MS | 20% | 2-5µg L$^{-1}$ |
| OM/OC ratio | SP-AMS | 6% (Aiken et al., 2008) | - |
| WSOA | HR-AMS,TOC | 6.9-8.5% | - |

2) Section 2.1 "Sampling site and PM2.5 collection": the authors need to mention here the artifacts related to the filter samplings, in particular the evaporation of semi-volatile compounds during the sampling. This is particularly important for some results presented later, such as the NO3-/SO42- ratio. Indeed, if these species are present under the form of ammonium nitrate and ammonium sulfate (as shown in section 3.2), ammonium nitrate will evaporate faster than ammonium sulfate during the sampling. Therefore, the concentrations of nitrate correspond to a lower limit, the real concentrations should be higher, and the real NO3-/SO42- ratios should also be higher. Another artifact concerns the adsorption of gases, such as volatile organic compounds (VOCs), onto the sampling media and collected particles, which can have an impact on the concentration of particle-bound polycyclic aromatic hydrocarbons (PAHs).

**Authors' reply:** Thanks for your suggestions. In the revised manuscript Sec. 2.1, we added the description that "Note filter-based measurements are inevitably subjected to various sampling artifacts including evaporation of semi-volatile species, and absorption of gases. Nitrate in the form of ammonium nitrate may have some evaporation loss as it is sensitive to temperature variations during sampling, and absorption of gases may influence the quantification of particle-bound polycyclic aromatic hydrocarbons (PAHs)."

3) Section 2.1 "Sampling site and PM2.5 collection": according to the wind rose plots presented in Fig. S1, the sampling site was under the influence of different air masses, depending on the season. This important information is not discussed in the section 3 "Results and discussion" and the corresponding sub-sections. Did the authors perform a back trajectory analysis to check where the air masses come from during each sampling period?

**Authors' reply:** That is a good suggestion. Now in Sec. 3.7, we performed back trajectory clustering analysis using the Hybrid Single-particle Lagrangian Intergrated trajectory (HYSPLIT) model. And relevant discussions were now added in the revised manuscript.

4) Section 2.2.5 "Offline SP-AMS analysis": it would be interesting if the authors explain here the advantages to use the SP-AMS for off-line analysis of filter samples. This kind of analysis presents several problems compared to on-line measurements: a) it has a much lower time resolution (20 hours in this study, instead of a few minutes), b) the total concentrations and size distributions of the species cannot be directly measured (given that the water extracts must be atomized), and c) it introduces artifacts related to the filter sampling. So what are the advantages of using that instrument for off-line analysis?

**Authors' reply:** In the first paragraph in Sec. 2.2.5, we added one paragraph. "The Aerodyne AMS is specially designed for online and real-time measurements of the submicron aerosol particles. The instrument has a very fine time resolution thus is powerful in capturing the quick atmospheric processes occurred in real atmosphere. While in this study, we used the SP-AMS for offline filter sample analyses. Compared with the online measurements, there are a few advantages: 1) it can greatly expand the application of AMS because it is often unrealistic to deploy the AMS for very long periods as it requires highly skilled personal to carefully maintain and operate the instrument; 2) for some sites, it is not accessible or not suitable for AMS deployment; 3) AMS analysis of organics can provide more details, for instance the elemental composition, oxidation states, etc., thus can offer useful insights into the origin of OA; 3) offline analyses may introduce artifacts compared with the online measurements,

but on the other hand, it also expands the size range as online measurements were often limited in submicron meter range."

5) Section 2.4 "Source apportionment of WSOA": the authors should say a few words on the robustness of this PMF analysis, given that the dataset contains only 69 samples (67 if the authors discarded two outliers) and corresponds to 20-hours averaged samples.

**Authors' reply:** Thanks for the suggestion. Regarding the number of samples used in PMF, we agree with the reviewer that in general, inclusion of more samples will provide better PMF results and more scientifically sound interpretation of the sources. Due to practical limitations, unfortunately we were only able to include 67 samples in the PMF calculation. Nevertheless, applications of PMF model on a limited number of samples were also reported previously, and can also provide very valuable insights into the sources and processes of the aerosols. For example, Huang et al. (2014) analyzed about in total 57 samples from 4 cities, and identified different sources during heavy haze formation in China; Sun et al. (2011) analyzed in total 24 samples from 4 sites to elucidate the sources and transformation processes of the water-soluble organic aerosols. In our case, we think our PMF analyses may still be trustworthy and valid, as the identified sources were reasonable as discussed later. Nevertheless, we have added relevant description in the manuscript regarding this caveat raised by the reviewer.

6) Lines 369-372: in addition to cations not measured by ion chromatography, a $NH_4^+$ measured/$NH_4^+$ predicted of 0.75 in winter can also simply be due to the presence of acids.

**Authors' reply:** In fact, as we have calculated the ion balance by using all measured ionic species, this sentence seems to be redundant, we now deleted it.

7) Lines 380-383: in summer, the high temperature may lead to a faster evaporation (not dissociation) of nitrate during the filter sampling. This may explain the lower NO3-/SO42- ratio in summer.

**Authors' reply:** We replace "dissociation" by "evaporation" in the revised manuscript.

8) Lines 389-391: the authors mention that the sulfur oxidation ratio was higher in summer. However, according to Fig. 6c, the difference with the other seasons does not seem significant.

**Authors' reply:** Yes, the SOR values in summer were not always higher than those in other seasons. But on average and statistically, it is indeed a bit higher. We have change the sentence to be more accurate "On average, the SOR value appears to be a bit higher in summer…"

9) Lines 538-542: it is surprising to notice that the O/C ratio of organics remained almost constant during the four seasons (Fig. 2), while we could expect higher values in summer due to increased photochemical activities. Can the authors say a few words on this in the manuscript? Among all the results presented in this manuscript (OC/EC ratio, etc.), only a higher sulfur oxidation ratio in summer seems to show increased photochemical activities during that period.

**Authors' reply:** Our measurement results do show similar O/C ratios of the organics across different seasons. It is likely due to: 1) we determined the water-soluble fraction of OA and the WSOA is typically the fraction with higher oxidation states in all seasons; 2) during summer, the gas-phase oxidation may be enhanced, while in other seasons, other oxidation pathway may dominate (likely aqueous-phase pathway), thus on average, the ambient OA yields similar O/C ratios among different seasons. This also indicates that more investigations are necessary to elucidate the detailed formation mechanism and evolution of OA in this region.

**Technical comments:**

10) Several correlation coefficients are reported throughout the manuscript. Sometimes, the authors use the Pearson's coefficient r, and sometimes the $r^2$. It will be better to be consistent and use systematically the same correlation coefficient, either r or $r^2$.

**Authors' reply:** We use systematically the same correlation coefficient Pearson's coefficient $r$ instead of $r^2$ in the revised manuscript.

11) When at least two references are given in parentheses, please add a space after the semicolons.

12) Title (also in the supplementary material): "Chemical characterization of fine

particular particulate matter".

13)Line 24: "the fine particular particulate matter (PM2.5) samples".

**Authors' reply:** Corrected.

14) Line 103: "short-term" is quite vague here. The typical duration of field campaigns with the AMS is approximately one month.

**Authors' reply:** we deleted the word "short-term".

15) Lines 183 and 446: the authors may mention in the title of these two sub-sections that they are talking about particle-bound PAHs, not about gas-phase PAHs.

**Authors' reply:** Yes, we have added such information as suggested.

16) Line 276: by which factor were ions with S/N ratios between 0.2 and 2 downweighed?

**Authors' reply:** The ions with S/N ratios between 0.2 and 2 were downweighted by a factor of 2.

17) Line 301: "Previous studies shows showed that low".

**Authors' reply:** Thanks, we have corrected this typo.

18) Line 363: actually, the NH4+ measured/NH4+ predicted ratio was first presented by Zhang et al. (2007), and used in tens of papers afterwards, Young et al. (2016) being one of them. Therefore, I would suggest to replace this reference.

**Authors' reply:** Thanks, we have used the original reference.

20) Figure 7: it would be important to include error bars corresponding to the standard deviations. This is particularly important for the Zn concentration in winter: is this high value due to 1-2 outliers, or do all the samples have a high value?

**Authors' reply:** Thanks. we added error bars corresponding to the standard deviations in Fig.7.

21) Figure 9: please scale the x-axes of the two panels the same way (either *m/z* 10-100 or 10-120).

**Authors' reply:** The x-axes are now all 10-100.

**References:**

Aiken, A. C., Decarlo, P. F., Kroll, J. H., Worsnop, D. R., Huffman, J. A., Docherty, K. S., Ulbrich, I. M., Mohr, C., Kimmel, J. R., Sueper, D., et al., O/C and OM/OC ratios of primary, secondary, and ambient organic aerosols with high-resolution time-of-flight aerosol mass spectrometry. Environ. Sci. Technol. 2008, 42 (12), 4478-4485.

Bozzetti, C., Sosedova, Y., Xiao, M., Daellenbach, K. R., Ulevicius, V., Dudoitis, V., Mordas, G., Byčenkienė, S., Plauškaitė, K., Vlachou, A., et al., Argon offline-AMS source apportionment of organic aerosol over yearly cycles for an urban, rural, and marine site in northern Europe. Atmos. Chem. Phys. 2017, 17 (1), 117-141.

Canagaratna, M. R., Jimenez, J. L., Kroll, J. H., Chen, Q., Kessler, S. H., Massoli, P., Hildebrandt Ruiz, L., Fortner, E., Williams, L. R., Wilson, K. R., et al., Elemental ratio measurements of organic compounds using aerosol mass spectrometry: characterization, improved calibration, and implications. Atmos. Chem. Phys. 2015, 15 (1), 253-272.

Huang, R., Zhang, Y., Bozzetti, C., Ho, K., Cao, J., Han, Y., Daellenbach, K. R., Slowik, J. G., Platt, S. M., Canonaco, F., et al., High secondary aerosol contribution to particulate pollution during haze events in China. Nature 2014, 514, 218-222.

Liang, C., Duan, F., He, K., Ma, Y., Review on recent progress in observations, source identifications and countermeasures of PM2.5. Environ. Int. 2016, 86, 150-170.

Mirante, F., Salvador, P., Pio, C., Alves, C., Artiñano, B., Caseiro, A., and Revuelta, M. A.: Size fractionated aerosol composition at roadside and background environments in the Madrid urban atmosphere, Atmos. Res., 138, 278-292, 2014.

Xu, L., Guo, H., Weber, R. J., Ng, N. L., Chemical Characterization of Water-Soluble Organic Aerosol in Contrasting Rural and Urban Environments in the Southeastern United States. Environ Sci Technol 2017, 51 (1), 78-88.